# The protease GtgE from Salmonella exclusively targets inactive Rab GTPases

Rudolf Wachtel [1], Bastian Bräuning [1], Sophie L. Mader [1], Felix Ecker[1], Ville R.I. Kaila [1], Michael Groll[1] & Aymelt Itzen [1]

Salmonella infections require the delivery of bacterial effectors into the host cell that alter the regulation of host defense mechanisms. The secreted cysteine protease GtgE from *S*. Typhimurium manipulates vesicular trafficking by modifying the Rab32 subfamily via cleaving the regulatory switch I region. Here we present a comprehensive biochemical, structural, and computational characterization of GtgE in complex with Rab32. Interestingly, GtgE solely processes the inactive GDP-bound GTPase. The crystal structure of the Rab32: GDP substrate in complex with the inactive mutant $GtgE_{C45A}$ reveals the molecular basis of substrate recognition. In combination with atomistic molecular dynamics simulations, the structural determinants for protein and activity-state specificity are identified. Mutations in a central interaction hub lead to loss of the strict GDP specificity. Our findings shed light on the sequence of host cell manipulation events during Salmonella infection and provide an explanation for the dependence on the co-secreted GTPase activating protein SopD2.

---

[1] Center for Integrated Protein Science Munich (CIPSM), Department Chemistry, Technical University of Munich, Lichtenbergstrasse 4, 85747 Garching, Germany. Rudolf Wachtel and Bastian Bräuning contributed equally to this work. Correspondence and requests for materials should be addressed to A.I. (email: aymelt.itzen@tum.de)

Central regulatory elements of intracellular signaling networks are small GTPases (guanosine triphosphate hydrolases, also referred to as G-proteins). G-proteins function as binary molecular switches and thereby inhibit or stimulate cellular networks via spatio-temporal protein–protein interactions[1]. In order to function in this capacity, the small GTPases can exist in the following two states: (1) they are inactive when complexed to guanosine diphosphate (GDP), and (2) they become active once bound to guanosine triphosphate (GTP). In the active state, signals are propagated by the recruitment of effector proteins that function as signal transmitting molecules. The activity states are regulated by dedicated regulatory enzymes that facilitate activation and inactivation of the G-proteins: guanine nucleotide exchange factors (GEFs) activate small GTPases by stimulating GDP-release and facilitating GTP-binding. In contrast, GTPase activating proteins (GAPs) accelerate inactivation via increasing the rate of intrinsic GTP-hydrolysis, converting the G-protein back to the inactive state. The activity states of small GTPases are manifested in specific conformations of two highly important regulatory loop regions that are referred to as switch I and switch II: the switch-regions are structurally disordered in the inactive state, but adopt a well-defined structure in the active form. The Rab-family of small GTPases is involved in regulating intracellular vesicular trafficking temporally and spatially[1]. These proteins are part of the so-called Rab-cycle that is characterized by a series of peripheral membrane association and cytosolic localization linked to nucleotide exchange. In order to attach to a membrane, Rab-proteins are post-translationally modified with one or two geranylgeranyl lipids at their C-termini. In the active state, these lipids confer strong membrane affinity and localize the Rab-protein to the cytosolic surface of intracellular compartments. In the inactive state, however, lipidated Rabs are targeted with high affinity by the GDP-dissociation inhibitor (GDI) that solubilizes the GTPase and thereby causes cytosolic localization.

Due to the pivotal role of small GTPases in intracellular signaling and the maintenance of cellular homeostasis, they are frequently targeted by bacterial pathogens in order to promote the infection[2]. An intriguing example in this respect is the bacterium *Salmonella enterica*. This bacterial pathogen is composed of a number of serovars that can infect diverse vertebrate species[3,4]. The *Salmonella enterica* Typhi (*S.* Typhi) causes typhoid fever by infecting only humans, whereas the *Salmonella enterica* Typhimurium (*S.* Typhimurium) can infect a broad host range. The molecular basis of this host range selection is not entirely clear. Recently, however, Galán and coworkers discovered that a protein encoded by the gene *gtgE* in *S.* Typhimurium contributes to host selection[5]. Consequently, expression of *gtgE* in *S.* Typhi counteracts host restriction and enables colonization of mice, thus, broadening host specificity. The protein GtgE has been identified as a cysteine protease that targets members of the Rab-family[5–8]. In particular, action of GtgE results in specific proteolytic cleavage of Rab29, Rab32, and Rab38 in vitro. The cleavage site is located in the regulatory switch I region of these Rabs and it takes place between G59 and V60 (amino acid numbering refer to Rab32)[6]. Rab32 is involved in controlling the biogenesis of lysosome-related organelles (LRO)[9–14]. It is believed that GtgE inactivates Rab32 proteolytically and thereby prevents the delivery of antimicrobial factors that would impair Salmonella infection. Since Salmonella is taken up by phagocytosis and evades destruction in the Salmonella-containing vacuole (SCV), which also constitutes an LRO, the Rab32-inactivating activity of GtgE may be an important component of the infectious mechanism[5].

Individual atomic structures of GtgE have been reported recently[7,8]. However, the molecular basis of the recognition of Rab32 by GtgE and the biochemical details of binding have not yet been addressed. Here we present the crystal structure of the Rab32:GtgE complex. Furthermore, by using a combination of biochemical characterization, molecular dynamics simulation, and binding studies, we demonstrate that GtgE shows exclusive selectivity for the inactive GDP-bound state of Rab32. The biochemical characterization of the GDP-state specificity of GtgE explains the previously reported necessity of Rab32:GTP inactivation by the secreted Salmonella GAP SopD2[15].

## Results

**GtgE specifically cleaves inactive, GDP-bound GTPases**. GtgE is a cysteine protease that cleaves the Rab32-subfamily comprising Rab29, Rab32, and Rab38, but not the closest related homolog Rab23[5]. Initially, we verified the previously observed Rab-specificity of GtgE (Fig. 1a). GtgE-mediated Rab-cleavage was monitored using denaturing sodium dodecylsulfate polyacrylamide gel electrophoresis (SDS-PAGE) on the basis of the molecular weight decrease by 6.5 kDa (Rab32, proteolysis at G59) or by 4.5 kDa (Rab29, Rab38, proteolysis at G41 and G43, respectively) (for sequence alignment of Rab32, Rab29, Rab38 and Rab23, Supplementary Fig. 1). Indeed, Rab29, Rab32, and Rab38, but not Rab23, are cleaved in the presence of catalytic quantities of GtgE.

We next asked whether GtgE is selective for Rab-proteins in the active (GTP) or inactive (GDP) state. To this end, the Rab-proteins were prepared in defined nucleotide states with GDP, GTP, or the non-hydrolyzable GTP-analog GppNHp. SDS-PAGE based monitoring of GtgE-mediated Rab-cleavage indicates that only the inactive GDP-bound, but not the active GppNHp-loaded GTPases, are processed by GtgE (Fig. 1b). Additionally, active Rab32 was purified from expression in *Escherichia coli* bound to GTP to ~90% (Supplementary Fig. 2A). Upon incubation with GtgE, Rab32:GTP was proteolytically cleaved significantly slower than the GDP-loaded form (Fig. 1b and Supplementary Fig. 2B). In order to investigate whether Rab32:GTP or the Rab32:GDP produced via intrinsic GTP-hydrolysis are the target of GtgE, we separately quantified GtgE-mediated Rab32:GTP-proteolysis and GTP-hydrolysis. Rab32 has an exceptionally slow intrinsic GTPase-activity with an hydrolysis constant determined to $k_{hydr} = 6.5 \cdot 10^{-6}\,s^{-1}$ (Fig. 1c, Supplementary Fig. 2A). The rate of Rab32:GTP-proteolysis in the presence of GtgE is identical to the rate of GTP-hydrolysis with an observed rate constant of $k_{obs,prot} = 6.7 \cdot 10^{-6}\,s^{-1}$ (Fig. 1c, Supplementary Fig. 2B). This demonstrates that the proteolytic turnover of Rab32:GTP is limited by the rate of intrinsic GTP-hydrolysis. Consequently, only the inactive (GDP-bound), but not the active (GTP-bound) Rab, is the target of GtgE.

We therefore wondered whether GtgE cooperates with other enzymes in order to convert the Rab-proteins into the inactive form as required for proteolysis. Interestingly, Salmonella secretes the bacterial Rab32-GAP SopD2 that produces Rab32:GDP from Rab32:GTP[15]. Indeed, proteolysis only happens in vitro when SopD2 is added catalytically to the reaction between Rab32:GTP and GtgE (Supplementary Fig. 3). The rate of GtgE-mediated cleavage for Rab32:GTP in the presence of SopD2 was indistinguishable from the reaction of Rab32:GDP in the absence of this GAP. This indicates a cooperative deactivation mechanism of SopD2 with GtgE for effective Rab32-cleavage.

We further note that Rab32:GDP-proteolysis in vitro does not precipitate the protein although the cleavage site is located in the structurally and functionally crucial switch I region (Supplementary Fig. 4). In fact, GtgE-cleaved Rab32:GDP could be preparatively produced and purified by size-exclusion chromatography. The protein elutes indistinguishably from the uncleaved

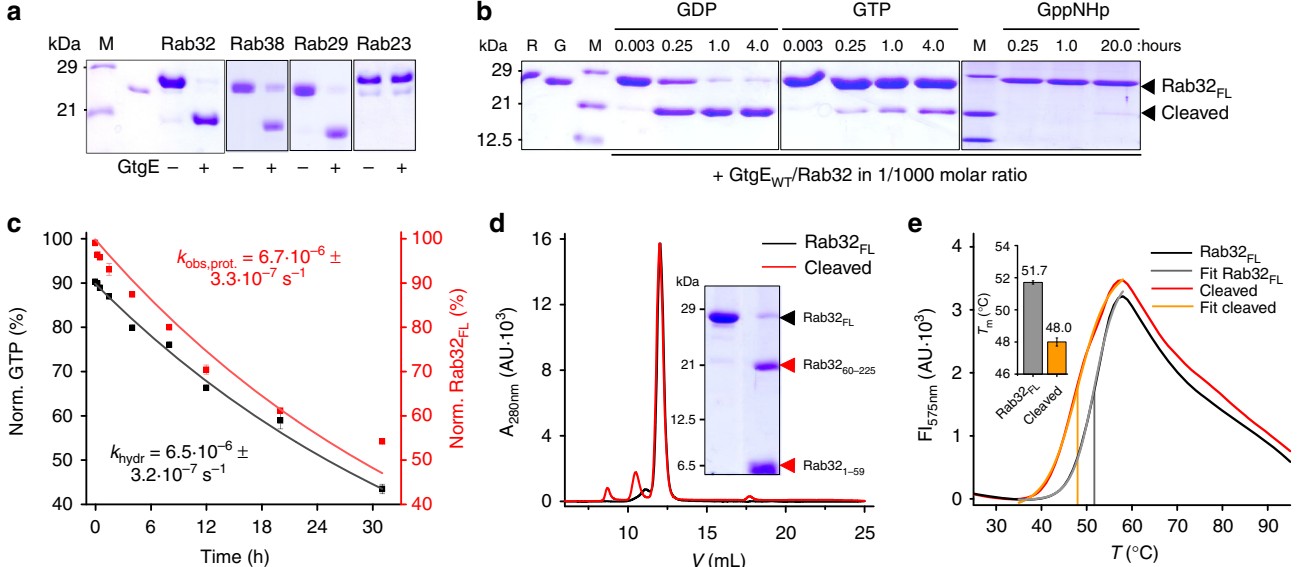

**Fig. 1** GtgE specifically cleaves the inactive form of Rab32-family members. **a** Substrate range of GtgE is restricted to Rab32 family members. Proteolytic gel shift assay of the Rab32-subfamily and Rab23. **b** Nucleotide-dependent GtgE-mediated (G, 8 nM) cleavage of $Rab32_{FL}$ (R, 8 μM) bound to GDP, GTP or GppNHp in a time-dependent SDS-PAGE gel shift assay. **c** Direct comparison of the intrinsic GTP-hydrolysis of Rab32:GTP (black) and the GtgE-mediated proteolysis of Rab32:GTP (red). GTP-hydrolysis was monitored by quantifying the GTP-content using ion-pairing reversed-phase chromatography (Supplementary Fig. 2A). The proteolysis was monitored by Coomassie-stained SDS-PAGE (Supplementary Fig. 2B). Both data sets are presented as means of technical replicates ($\pm$ SEM; $n = 2$). The data were fitted to a single exponential function, yielding the observed rate constants $k_{obs,prot}$ and $k_{hydr}$. **d** Proteolytic cleavage does not promote unfolding of Rab32. Comparison of the retention times of $Rab32_{FL}$ (8 μM, black) and cleaved Rab32 (8 μM, red) by size-exclusion chromatography (SEC). Inset: Full conversion was verified by SDS-PAGE. **e** Rab32 cleavage mildly affects thermal stability. Melting point ($T_m$) determination of $Rab32_{FL}$ (black) and cleaved Rab32 (red) by differential scanning fluorimetry (DSF). The data were fitted to a Boltzmann equation, yielding the melting temperature. Inset: Results are presented as bar graphs with means $\pm$ SEM ($n = 3$)

form (Fig. 1d). We thus probed whether switch I cleavage could affect the folding stability of the G-protein. Rab32:GDP and GtgE-cleaved Rab32:GDP were subjected to thermal unfolding experiments using differential scanning fluorimetry (DSF) (Fig. 1e). The melting points ($T_m$) were determined from fitting the fluorescence data to Boltzmann equation to $T_{m,FL} = 52\,°C$ and $T_{m,cleaved} = 48\,°C$ for uncleaved and cleaved Rab32:GDP, thus revealing that cleaved Rab32:GDP behaves like a well-folded and stable protein. In summary, we have shown that GtgE demonstrates exclusive specificity for inactive GDP-bound Rab-proteins, and that the cleaved Rab32 remains a stable monomeric protein.

**GtgE has high affinity for inactive Rab substrates**. We next addressed the enzymatic properties of GtgE for inactive Rab32:GDP. First, time-resolved kinetics of GtgE-mediated Rab32:GDP cleavage were performed. We processed Rab32:GDP with catalytic quantities of GtgE, separated heat-inactivated samples from different reaction time points on the SDS-PAGE, and quantified Rab32-cleavage educts by densitometric quantification of Coomassie-stained bands since traditional fluorescence-based approaches did not function in this case (Fig. 2a, b and see Supplementary Fig. 5A, B). The catalytic efficiency ($k_{cat}/K_m$) of GtgE was determined to $5·10^5\,M^{-1}\,s^{-1}$. For this purpose, the observed rate constant ($k_{obs}$) was obtained from fitting the dependence of $Rab32_{FL}$-cleavage over time to a single exponential curve (Fig. 2b), followed by division of $k_{obs}$ by the GtgE concentration. The obtained $k_{cat}/K_m$ value for GtgE implies a high catalytic activity, and suggests that GtgE operates at the maximum proteolysis rate even at low micromolar concentrations of Rab32:GDP. This property is indicative of a high affinity enzyme-substrate-complex having a Michaelis–Menten constant

($K_M$) in the micro- or nanomolar range. Determination of the $K_M$ would require Michaelis–Menten-kinetics to be conducted at micromolar to submicromolar Rab32:GDP-concentrations. However, at such low substrate concentration, the quantification of cleavage educts and products by Coomassie-stained SDS-PAGE gels would be challenging. As an alternative, we aimed to determine the dissociation equilibrium constant ($K_D$) between Rab32:GDP and a catalytically inactive GtgE-mutant as a surrogate for $K_M$. The amino acid residue C45 has previously been shown to be important for the proteolytic activity of GtgE, and likely constitutes the active site nucleophile of this cysteine protease[7,8]. We therefore confirmed that $GtgE_{C45A}$ indeed has no proteolytic activity on Rab32:GDP (Supplementary Fig. 6), allowing us to determine the $K_D$ by two independent approaches. First, we subjected Rab32:GDP (20 μM) to isothermal titration calorimetry (ITC) with $GtgE_{C45A}$ (200 μM), revealing a dissociation constant of $K_{D,ITC} = 96 \pm 31\,nM$ (Fig. 2c). The determined stoichiometry confirms that GtgE and Rab32:GDP form a heterodimeric 1:1-complex. In parallel, we utilized analytical ultracentrifugation (aUC) of Rab32:GDP (200 nM) covalently labeled at thiol groups with Atto488-Maleimide ($Rab32:GDP_{fluor.}$), and quantified complex formation with increasing concentrations of $GtgE_{C45A}$ by the difference in $Rab32:GDP_{fluor.}$ sedimentation (Fig. 2d). Assuming a 1:1 binding stoichiometry, the $K_D$ of the interaction is determined to $K_{D,aUC} = 118 \pm 29\,nM$ which is in good agreement with the ITC-experiment. Thus, our orthogonal approaches determined the $K_D$ for the interaction between Rab32:GDP and the inactive $GtgE_{C45A}$ mutant to be approximately 100 nM. Assuming that the $K_D$ equals the Michaelis–Menten constant $K_m$ for the enzymatic process, we obtain a turnover rate($k_{cat}$) of $k_{cat} = 0.05\,s^{-1}$ by multiplying $k_{cat}/K_m$ (Rab32:GDP, GtgE) $= 5.1·10^5\,M^{-1}\,s^{-1}$ with the $K_D$. In summary, as indicated by a low $K_D$ and a moderate $k_{cat}$, our

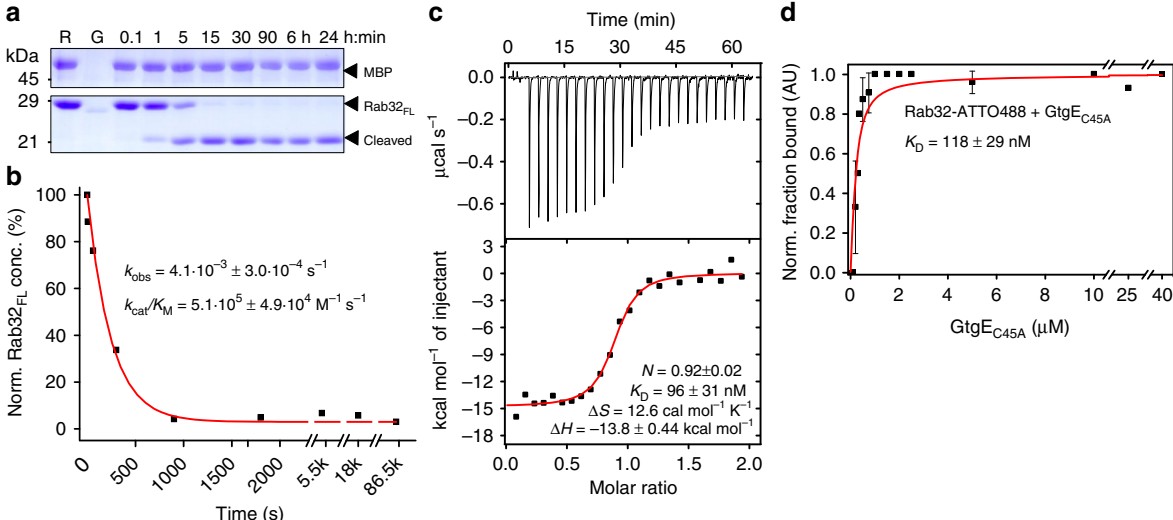

**Fig. 2** Inactive mutant GtgE$_{C45A}$ has high affinity for Rab32:GDP. **a** Time-dependent GtgE-mediated cleavage kinetic of Rab32:GDP with a SDS-PAGE gel shift assay. R Rab32$_{FL}$ (8 µM); G: GtgE$_{WT}$ (8 nM). Spiked maltose binding protein (MBP, 2 µM) was used as internal quantification reference. **b** Densitometric analysis of Rab32$_{FL}$ bands from panel **a** fitted to single exponential function. The rate constant of the fit divided by the enzyme concentration yields the catalytic efficiency for GtgE. **c** ITC of Rab32$_{FL}$:GDP (20 µM) titrated with 200 µM GtgE$_{C45A}$. Integrated heat peaks were fitted to a one-site-binding model yielding the binding stoichiometry ($N$), the enthalpy ($\Delta H$), the entropy ($\Delta S$), and the dissociation constant ($K_D$). The data are presented as means ± SEM ($n = 3$). **d** Analytical ultracentrifugation (aUC) experiments of the binding affinity from fluorescently labeled Rab32$_{FL}$:GDP titrated with GtgE$_{C45A}$. The normalized fraction of bound Rab32 was fitted to a hyperbolic equation, yielding the $K_D$. The data are presented as means ± SEM ($n = 2$)

findings indicate that GtgE is a catalytically efficient enzyme with a high substrate affinity.

**The crystal structure of the Rab32:GDP:GtgE complex**. The high affinity of GtgE$_{C45A}$ for Rab32:GDP allowed us to purify thecomplex (Supplementary Fig. 7). Crystals obtained after reductively alkylating the truncated Rab32$_{18-201}$:GDP:GtgE$_{21-214, C45A}$-complex and the full-length complex Rab32$_{1-225}$:GDP:GtgE$_{1-228,C45A}$ diffracted up to 2.3 Å and 2.9 Å resolution, respectively (Supplementary Fig. 8)[16]. The X-ray structures of the complexes were solved by the molecular replacement technique using the coordinates of Rab32:GppCH$_2$p (PDB ID: 4CYM[17]) and truncated GtgE (PDB ID: 4MI7[7]) as appropriate search models (Fig. 3a) (see Supplementary Table 1 for data collection and refinement statistics). The average B-factor of the full-length complex is higher than for the truncated complex, though the entire electron density is well defined for both structures (Supplementary Fig. 9). Due to the higher resolution, the truncated Rab32$_{18-201}$:GDP:GtgE$_{21-214,C45A}$-complex is discussed only and referred to as Rab32:GDP:GtgE$_{C45A}$ hereafter.

The complex shows the general features of the previously determined individual subunits of Rab32 and GtgE[7,8,17]. The Cα-atoms of the Rab32-molecules superimpose for 121 amino acids (excluding the structurally-flexible switch-regions, switch I$_{aa48-61}$ and switch II$_{aa82-99}$) with an RMSD of 0.52 Å[18]. In case of GtgE, 97 Cαs were aligned with an RMSD of 0.65 Å (excluding the previously unresolved amino acids 23–54). Rab32 is composed of a central β-sheet comprising six β-strands (β$_{1R}$–β$_{6R}$, where subscript R denotes Rab32) and five α-helices (α$_{1R}$-α$_{5R}$) (Fig. 3a, b; Supplementary Fig. 1). However, no electron density is observed for the helix α$_{5R}$, which is usually present in all small GTPases. GtgE is composed of a six-membered central antiparallel β-sheet that is perpendicularly arranged in a tip-to-tip manner to the β-sheet of Rab32. The GtgE-β-sheet is flanked by five α-helices of which the two N-terminal α-helices and the three C-terminal helices are on opposite ends of the sheet structure (Fig. 3c)[8]. The amino acid regions 23–44 and 60–82

appear to be folded within the active enzyme structure only after complex formation with Rab32, as these peptide stretches occupy significantly different positions in comparison to the GtgE-apo-structure (Fig. 3c).

In the complex structure, GtgE can be envisioned as a trident poking into Rab32 with three spike-like protrusions (Fig. 3a, d). Spike 1 comprises amino acid region 38–45 in conjunction with region 76–82. Spike 2 is represented by the β$_{2G}$–β$_{3G}$ loop (where subscript G denotes GtgE), and penetrates deeply into the switch I–switch II cleft of Rab32. Thereby, spike 1 and 2 form intimate interactions with the switch I structure that is also the site of proteolytic cleavage. Spike 3 is formed by the α$_{4G}$-helix and a C-terminal loop extension. The shallow cleft between spike 2 and 3 is filled by the switch II region of Rab32.

Binding of GtgE to Rab32 does not result in major structural changes outside of the switch-regions when compared to the structure of Rab32:GppCH$_2$p (Fig. 3b)[17]. Since GtgE is specific for the inactive GDP-bound Rab32 (Fig. 1), this suggests that the contacts within switch-regions undergo large-scale conformational changes upon GtgE-binding. The switches are reminiscent of, but not identical to the conformations observed in Rab-proteins in complex with the GDP-state-specific interaction partner GDI[19,20] (Supplementary Fig. 10). Spike 2 of GtgE acts like a wedge that drives the switch-regions further apart than observed in Rab:GDI-complexes. On the other hand, binding of Rab32 to GtgE does not affect the secondary structure elements of the protease (Fig. 3c)[7,8]. Rather, flexible loops previously unrecognized inthe GtgE-structure, become ordered in the Rab32:GtgE complex. In particular, the N-terminus (amino acids 23–45), including the position of the active sitecysteine (here the C45A substitution) is now resolved, in addition to the loop regions β$_{2G}$-β$_{3G}$ (i.e., spike 2) and α$_{4G}$-β$_{6G}$ (i.e., spike 3).

The complex interface buries an average solvent exposed surface area of 1535 Å$^2$, as determined with the 'Protein interfaces, surfaces and assemblies' web service (PISA)[21]. It is mainly composed of the switch- and β$_{2R}$-β$_{3R}$-regions of Rab32 and the three spikes of the GtgE-trident (Fig. 3d, e). A large number of hydrophobic and polar interactions contribute to the

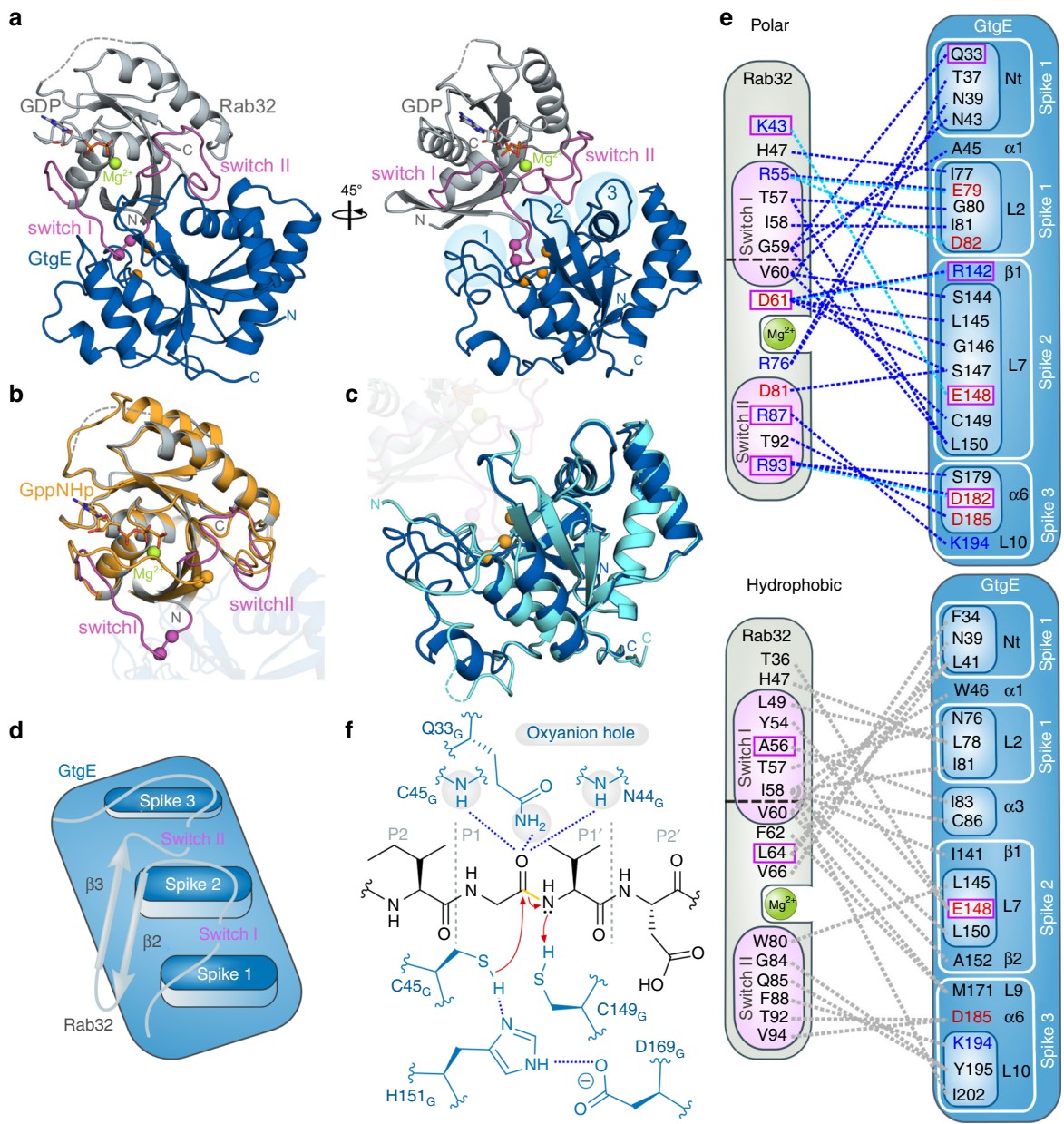

**Fig. 3** Rab32:GDP:GtgE$_{C45A}$ complex structure. **a** Cartoon depiction of the Rab32:GDP:GtgE$_{C45A}$-complex. Sticks: GDP; magenta: switch regions; magenta spheres: cleavage site, green sphere: Mg$^{2+}$-ion; orange spheres: catalytic triad; light blue circles: spike 1–3. **b** Structural superposition of Rab32:GppCH$_2$p (PDB ID: 4CYM, wheat[17]) and Rab32:GDP:GtgE$_{C45A}$ (gray). Transparent: GtgE structure. For labeling see panel **a**. **c** Structural superposition of GtgE (PDB ID: 5KDG, light blue[8]) and Rab32:GDP:GtgE$_{C45A}$.(blue). Transparent: Rab32:GDP structure. For labeling see panel **a**. **d** Schematic model of the Rab32:GtgE-binding interface, including the Rab32 switch regions and the adjacent secondary structure elements. The switch regions are positioned into the cavities established by GtgEs spike 1/spike 2 and spike 2/spike 3. **e** Schematic representation of all polar (top) and hydrophobic interactions (bottom). Acidic and basic amino acids are depicted in red and blue, respectively (Supplementary Table 2). Interactions highlighted in dashed lines: hydrogen bonds (blue), salt-bridges (cyan blue), hydrophobic (gray). L: loop, α: α-helix, β: β-sheet (Supplementary Fig. 1). Magenta box: residues tested by mutations in a GtgE-mediated cleavage assay (Fig. 4c, d). **f** Schematic of the catalytic mechanism of GtgE. Cleavage occurs in switch I between G59$_R$ (P1) and V60$_R$ (P1′) by the catalytic triad (C45$_G$, H151$_G$, D169$_G$). Q33$_G$, C45$_G$, N44$_G$: oxyanion hole; proton donor: C149$_G$. Blue: GtgE amino acids, yellow: cleaved bond, bluedashed lines: polar interactions. P1, P2, P1′, and P2′ correspond to the Schechter-Berger-nomenclature of protease substrates

complex interface, providing a potential molecular basis for the observed high affinity between GtgE and Rab32:GDP (Fig. 3e). In particular, the Rab32 residues adjacent to the cleavage site between G59$_R$ and V60$_R$ are intimately involved in interacting with the deep spike 1/spike 2 cavity of GtgE. The switch I residue I58$_R$ forms the center of a hydrophobic cluster and interacts with W46$_G$, L81$_G$, C86$_G$, I141$_G$, L150$_G$, and I202$_G$, whereas D61$_R$ from the protease recognition sequence forms a specific interaction

with the guanidinium group of R142$_G$. Three further ionic interactions K42$_R$-E148$_G$, R55$_R$-D82$_G$, and R93$_R$-D182$_G$ are likely to contribute to binding and enhancing specificity.

The interacting polar network between the catalytic triad, C45$_G$, H151$_G$, and D169$_G$, is formed in the Rab32:GDP: GtgE$_{C45A}$-complex once C45$_G$ is modeled to the active site cysteine nucleophile (Supplementary Fig. 11)[6–8]. Furthermore, the oxyanion hole can be unambiguously assigned to the peptide

backbone NH-groups of $C45_G$ and $N44_G$ as well as the side chain $Q33_G$ $N^{\varepsilon2}$. (Fig. 3f), while the thiol-group of $C149_G$ potentially acts as the Brønsted acid to generate a proper N-terminal leaving group of $V60_R$ (Fig. 3f).

In summary, GtgE interacts tightly with Rab32 via three binding platforms referred to as spike 1–3 that accommodate the switch regions. In particular, the resolved structure reveals that the cleavage site of Rab32 is pulled into a deep cavity formed by spike 1 and 2.

**Confirmation of the interface of the Rab32:GDP:GtgE complex.** The crystal structure of Rab32:GDP:GtgE$_{C45A}$-complex suggests that many side chains within the switch region contribute to the interaction between the GTPase and the protease. In order to verify the significance of individual amino acids in complex formation, we investigated Rab32 point mutants for the ability to form a complex with GtgE by yeast-two-hybrid (Y2H) experiments using the full-length proteins (see Fig. 3e for positions of mutants). We first established an Y2H-based interaction assay by combining the proteolytically inactive mutant $C45A_G$ of GtgE as prey and the Rab32-subfamily together with Rab23 as bait (Fig. 4a). Complex formation is detected by growth on histidine-lacking media only for the $C45A_G$ mutant, but not the wild-type GtgE, indicating an interaction exclusively between the GTPase substrates and the proteolytically inactive GtgE. Consistent with our biophysical analyses we observe that this interaction is of high

affinity for Rab32 since growth was also observed under more stringent conditions in the presence of 10 mM 3′-AT (Fig. 4a). The interaction with Rab38 is also detected, albeit the positive signal is more susceptible to the presence of 3′-AT, indicating a weaker interaction. Rab29 is self-activating in the Y2H assay (left panel) and thus cannot be used for monitoring GtgE-binding.

Next, a library of Rab32 switch-region mutants was explored by Y2H-based interaction studies in order to verify the interface observed in the complex crystal structure. In the switch I region, the amino acid mutants $T52A_R$ (P3), $I58_R$ (P2), $G59_R$ (P1), $D61_R$ (P2′), and $F62_R$ (P3′) show a strong growth defect, confirming that they are involved in forming the Rab32:GDP:GtgE$_{C45A}$-complex (denominations in parentheses correspond to the Schechter-Berger-nomenclature for positions in the protease substrate cleavage site) (Fig. 4b, top). In particular, mutation of $I58_R$, identified as a hub for hydrophobic interaction in the crystal structure, does not support growth in Y2H, thereby supporting the complex interface and the significance of this residue.

We also conducted a Rab32 switch II mutational analysis by employing the Y2H-screening (Fig. 4b, bottom). However, none of the mutants directly involved in the complex interface demonstrate a detectable growth defect, indicating that switch II mutations may be tolerated due to the high complex affinity. In contrast, the mutants $W80A_R$ and $D81A_R$ in the interswitch-region show severe growth inhibition. The side chain of $W80_R$ is a constituent of the important Rab hydrophobic triad that is relevant for binding to most interaction partners[22]. Since the

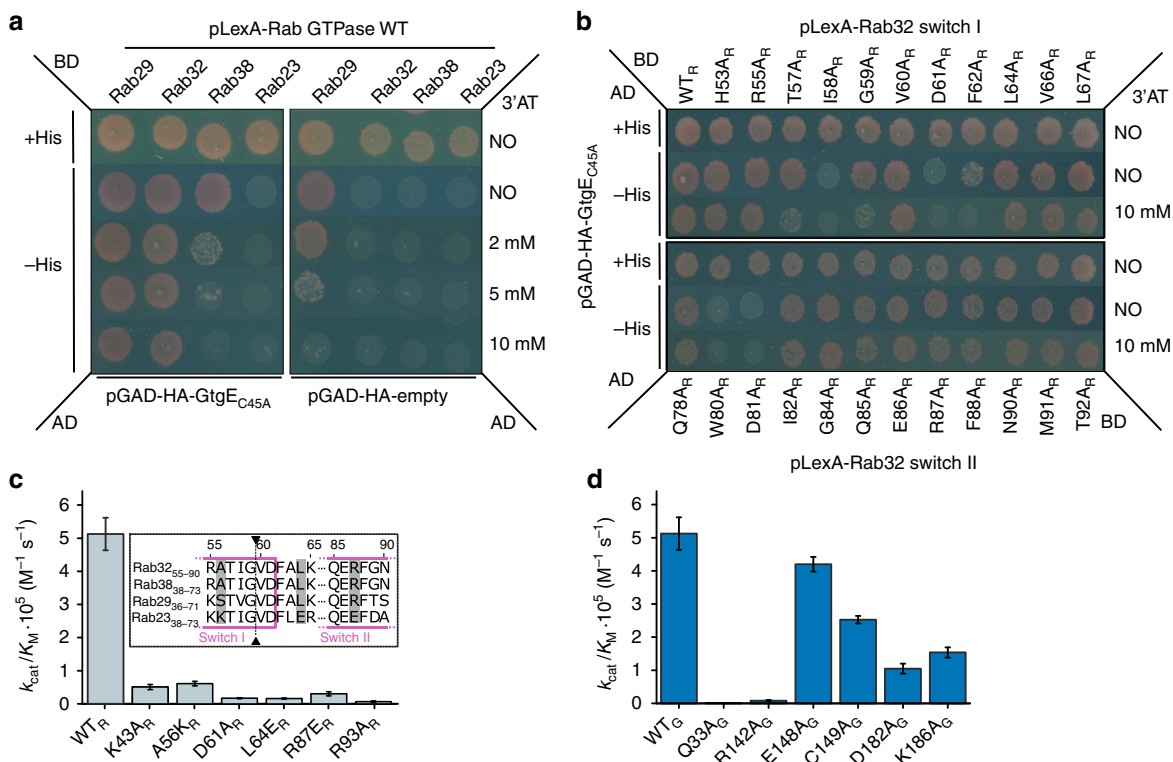

**Fig. 4** Analysis of the Rab32:GtgE complex interface and catalytic mechanism. **a** Y2H-assay with His growth selection for GtgE$_{C45A}$ (pGAD-HA; prey) binding to full-length Rab29, 32, 38 or Rab23 (pLexA; bait). For probing interaction strength, the growth inhibitor 3-Aminotriazol (3′-AT) was titrated from 0–10 mM. **b** Mutational alanine screening of the Rab32 switch I (top) and switch II region (bottom) with the Y2H assay. The switch I residues at the GtgE cleavage site display a significant growth defect upon addition of 3′-AT, supporting the Rab32:GDP:GtgE$_{C45A}$ interface. Also, the switch II residues $W80_R$ and $D81_R$ show decreased complex formation. **c** Cleavage efficiency of structure and sequence guided Rab32-mutants (8 µM) by GtgE (8 nM) obtained from a densitometric analysis SDS-PAGE-based activity assays (Supplementary Figs. 12A, 13A, Fig. 3e; and Supplementary Table 3). Inset: Partial sequence alignment of the Rab32 subfamily and Rab23. Gray: Residues mutated to the corresponding Rab23 moiety; arrows: GtgE cleavage site (see also Supplementary Fig. 1). **d** Catalytic efficiency of GtgE mutants with Rab32 wild type. Identical experimental setup as in panel **c** (Fig. 3e, Supplementary Figs. 12B, 13B, and Supplementary Table 3)

$W80_R$ side chain interacts with $F62_R$, the $W80A_R$ mediated growth defect can be explained by an influence on the positioning of the switch I $F62_R$ side chain that is important for GtgE binding (Fig. 4b, top).

In addition to the Y2H-assays, we probed the structure-activity relationship of the complex with mutants in all interacting regions (Rab switch-regions and GtgE-spikes, Fig. 3e) by studying GtgE-mediated Rab32-proteolysis (Fig. 4c, d; Supplementary Figs. 1, 13; Supplementary Table 3). All designed mutations in Rab32 were either constructed based on the structural analysis of the Rab32:GtgE complex ($K43A_R$, $D61A_R$, $R93A_R$) or inspired by the alignment between Rab32 and the non-GtgE substrate Rab23 ($A56K_R$, $L64E_R$, $R87E_R$) (Fig. 4c, Supplementary Fig. 1). The tested Rab32-mutants display at least a 10-fold activity reduction, supporting their significance for the substrate recognition and complex formation. Mutants $D61A_R$ and $L64E_R$ show a 30-fold decrease in activity with GtgE, emphasizing the importance of the salt bridge between $D61_R$ and $R142_G$. The $L64E_R$-mutation most likely induces a steric hindrance between $F34_G$ and $L41_G$, which further give insights into the Rab-substrate restriction. The mutants $K43A_R$ and $R93A_R$ abrogate salt bridge formation with $E148_G$ and $D182_G$, respectively, hence confirming the relevance of the corresponding ionic interaction for complex formation. In addition, the decrease in activity for the $R87E_R$ mutant supports the importance of the arginine side chain in binding to GtgE spike 3 via a backbone H-bridge with $K194_G$. The mutant $A56K_R$ presumably causes steric clashes with the side chains of $L150_G$ and $P143_G$ that form the basis of the important spike 2 loop of GtgE (Fig. 3e), thereby corroborating the relevance of this region for recognition of Rab32 by the protease.

Moreover, all GtgE-mutants except for $E148A_G$ show a significant drop in activity (Fig. 4d). In particular, $Q33A_G$ decreases the activity 200-fold, confirming its key role in the oxyanion hole. The importance of the $D61_R$-$R142_G$ salt bridge as a central interaction site mentioned before is once more confirmed using the $R142A_G$ mutation. The $C149A_G$ mutant reduces the activity by a factor of 2. Its localization near the active site suggests a role as a proton donor for the newly formed N-terminus during catalysis. The salt bridge formed by $D182_G$ with the switch II residue $R93_R$ shows a greater impact in the complex formation than the one constituted by $E148_G$ with $K43_R$. The residue $K186_G$ is indirectly involved in the complex interface, stabilizing spike 3 by forming two intramolecular polar contacts with $Q178_G$ and $E153_G$ near the catalytic base $H151_G$, rendering $K186_G$ an important organizing element in the switch II/spike 3-interface.

The Rab32:GDP:GtgE complex structure was also used to obtain insights into the Rab-selectivity. On the basis of the amino acid sequence alignment between the GtgE-substrates Rab32, 29, 38 and the non-GtgE-substrate Rab23, we identified four amino acids that are located in the protein–protein interface and may be incompatible with binding of Rab23 to the protease (Supplementary Fig. 1). Therefore, we tested whether Rab23 is converted into a GtgE-substrate by introducing the Rab32 analogous substitutions K40A, E48L, Q50V, and E70R in vitro. However, this Rab23 mutant was not cleaved by GtgE, indicating a multifactorial selection mode that cannot be reduced to a few specific amino acid side chains (Supplementary Fig. 14).

Thus, the mutational analysis confirms the Rab32–GtgE-interface observed in the complex crystal structure.

**Insights into the Rab32:GDP specificity of GtgE.** We next addressed the molecular basis for the strict preference of GtgE for the inactive GDP-bound Rab32 by analyzing the Rab:GDP:$GtgE_{C45A}$-complex crystal structure (Fig. 1b). Three potential hubs on Rab32 were identified that could hypothetically contribute to the targeting of GtgE to the inactive GTPase: hub 1 is constituted by the switch II residue $E86_R$, hub 2 by the switch I amino acid $Y54_R$, and hub 3 by the switch II side chain of $F88_R$ together with $K194_G$ and $E114_G$ (see below, Fig. 5a, b).

Close inspection of a structural superposition of the complex with active, $GppCH_2p$-bound Rab32 suggests a potentially important role of the hub 1 side chain of $E86_R$ in activity-state discrimination by GtgE[17]. In the GtgE-bound form, the side chain of $E86_R$ protrudes into a Rab32-cavity that would be occupied by the γ-phosphate of GTP or $GppCH_2p$ in the active state (Fig. 5a). Consequently, the conformational rearrangement of switch II by GtgE-binding would result in a steric and electrostatic clash of $E86_R$ with the γ-phosphate, thereby making binding of Rab32:GTP to GtgE incompatible. In order to test this hypothesis, we analyzed the GtgE-mediated proteolysis of the $E86A_R$-mutation of Rab32 with the full-length proteins. Surprisingly, the $E86A_R$-mutation did not change the activity-state preference of GtgE since only GDP-bound but not GppNHp-loaded Rab32-$E86A_R$ was a substrate as monitored by SDS-PAGE based proteolysis (Fig. 5c). Hence, additional factors contribute to the Rab32:GDP specificity of GtgE.

As a further potential decisive element of the nucleotide discrimination, we focused on the switch I amino acid $Y54_R$ as hub 2. The $Y54_R$-position is conserved among small GTPases and forms a polar interaction with the GTP γ-phosphate in the active state. In the Rab32:GDP:$GtgE_{C45A}$ structure, $Y54_R$ is turned away from the phosphates, pointing towards the guanine base. This observation could imply that $Y54_R$ is unable to support GTP-coordination in the Rab32:GtgE complex and therefore does not permit binding of GtgE to Rab32:GTP. Thus, the full-length mutants $Y54F_R$ and $Y54A_R$ were loaded with GDP or GppNHp and tested for their substrate properties towards $GtgE_{WT}$ (Fig. 5d and Supplementary Fig. 15A). Surprisingly, we find that again only the inactive GDP-bound Rab32-mutants were cleaved, and even after long incubation time no hydrolysis could be detected for the GppNHp-bound mutants, rendering the $Y54_R$ unimportant in the nucleotide-state selection.

In order to obtain additional insights into the nucleotide-state specificity of GtgE, we performed atomistic molecular dynamics (MD) simulations. We first considered calculating the structure of a hypothetical complex between active Rab32:GTP and GtgE. Therefore, the only currently available structure of active Rab32 bound to the non-hydrolysable GTP-analog $GppCH_2p$ was taken from its complex with the effector protein VARP (PDB ID: 4CYM)[17]. Replacement of the structure of the $GppCH_2p$-bound Rab32 from its complex with VARP resulted in steric clashes of Rab32 with GtgE despite the global structure of Rab32 closely resemble each other in the two X-ray structures (Supplementary Fig. 16). Instead of probing the formation of Rab32:GTP:GtgE complex by extensive free energy calculations, we computationally replaced GDP with GTP in the nucleotide binding pocket of Rab32 and probed the structural determinants for GtgE-dissociation from the complex. This approach is justified since small GTPases can form complexes with their binding partners in both activity states albeit with differing affinities[23]. It is common, however, that protein–protein affinities are mainly determined by the rate of complex dissociation ($k_{off}$, $10^{-5}$–$10^3$ M$^{-1}$ s$^{-1}$) rather than by the second-order rate constant of complex association ($k_{on}$, $10^6$–$10^7$ M$^{-1}$ s$^{-1}$). The MD-simulation, therefore, mimics the initial molecular steps of GtgE dissociating from Rab32:GTP, and aims to probe determinants that would contribute most to dissociating the Rab32:GTP:GtgE complex once formed. Notably, these computations do not address the rate of complex association, since they would require extensive free energy simulations, and are outside the scope of the present study.

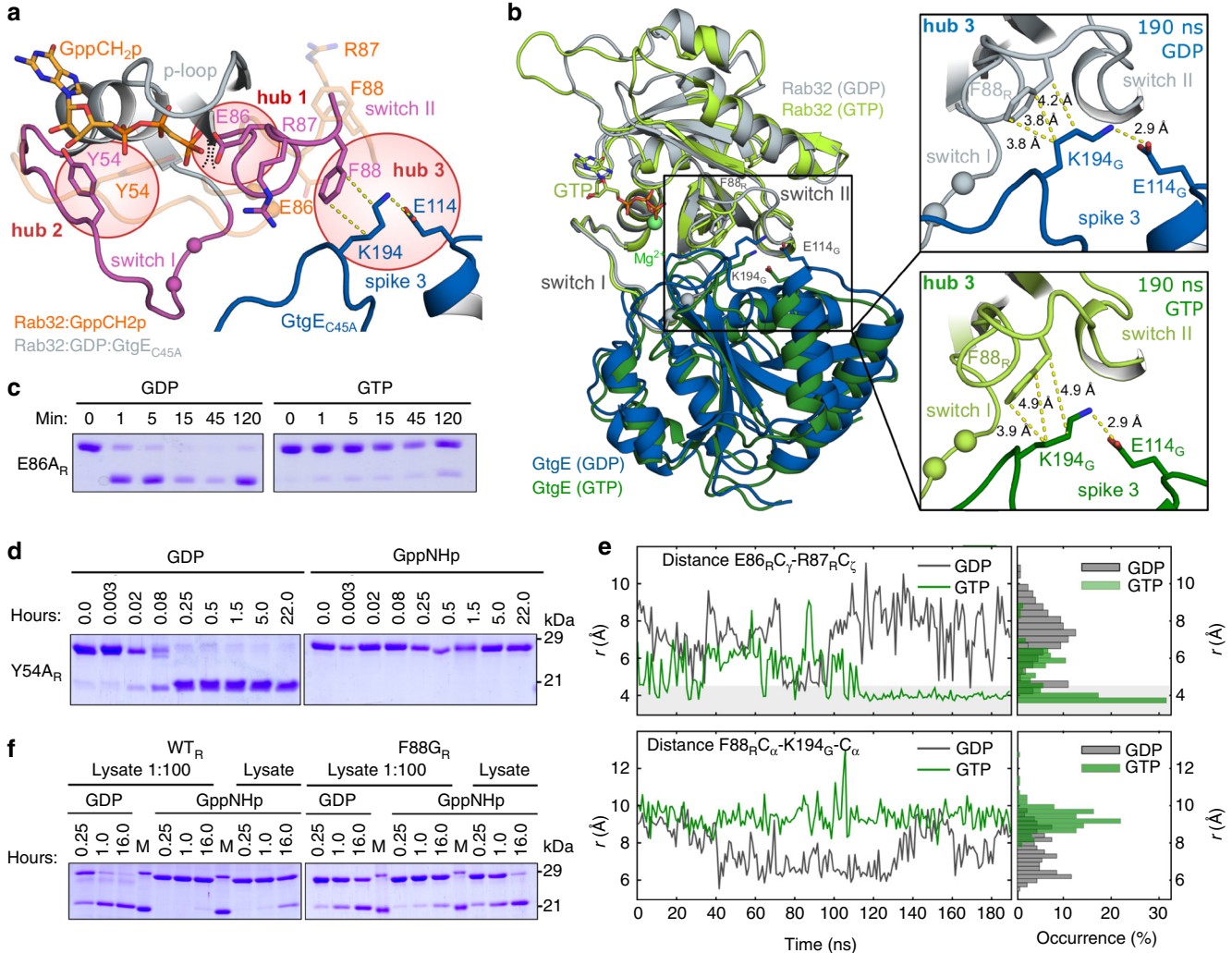

**Fig. 5** A molecular basis of the GDP-state specificity of GtgE. **a** Putative hubs involved in the GDP-state specificity of GtgE depicted in the superposition of active and inactive Rab32-structures. Gray: Rab32:GDP:GtgE$_{C45A}$; transparent wheat: Rab32:GppCH$_2$p:VARP (PDB ID: 4CYM[17]); sticks: Hub 1 (E86$_R$), hub 2 (Y54$_R$), hub 3 (comprising of F88$_R$, E114$_G$, and K194$_G$), and GppCH$_2$p; spheres: C$_\alpha$-atoms of the cleavage site; magenta: switch regions Rab32:GDP; black and yellow dashed lines: steric clash and hydrogen bonds, respectively. **b** Hub 3 (F88$_R$; E114$_G$; K194$_G$) of the Rab:GtgE$_{C45A}$ switch II interface. Structural superposition of MD-simulations of the Rab32:GtgE complex bound to either GDP (gray/blue) or GTP positioned in silico (light/dark green) after 190 ns. Right: van-der-Waals contacts of hub 3 indicated by yellow dashed lines for the GDP (top) and the GTP-complex (bottom), respectively (see Supplementary Fig. 18). **c, d** Processing of the hub mutants hub 1 E86A$_R$ (**c**), and hub 2 Y54A$_R$ (**d**) by GtgE in the GDP and GTP states, analyzed with a gel shift activity assay (see also Supplementary Fig. 15A). **e** Selected atom pair distances in a GDP-bound and hypothetical GTP-bound Rab32:GtgE complex from 190 ns MD-simulations (left panels) and histograms of respective distance occurrences (right panels), with an ion-pair distance threshold (< 4.5 Å) highlighted in gray. Top: E86$_R$ and R87$_R$ form a stable ion-pair after *ca.* 120 ns in the hypothetical GTP-bound complex (in green), initiated by an electrostatic repulsion from the γ-phosphate of GTP (see also Supplementary Fig. 18C). In contrast, no stable ion-pair between E86$_R$ and R87$_R$ is observed in the GDP-bound structure (in gray). Bottom: Distances between F88$_R$ of Rab32 and K194$_G$ of GtgE from MD-simulations of the GDP- (in gray) and GTP-bound (in green) Rab32:GtgE complexes. In the GDP-bound form, K194$_G$ forms an interaction with F88$_R$, while in the GTP-bound complex, K194$_G$ dissociates from F88$_R$, which may trigger the dissociation of GtgE from Rab32 (see also Supplementary Fig. 18A). The MD-simulations suggest that F88$_R$ may form a decisive element for the GDP-state preference of GtgE toward Rab32. **f** Mutations in Rab32 hub 3 change GtgEs nucleotide-state selectivity leading to cleavage of Rab32:GppNHp. Rab32 F88G loaded with GDP or GppNHp was treated with cleared *E. coli* lysate (or 1:100 diluted lysate) overexpressing wild-type GtgE and analyzed by an gel shift assay (see also Supplementary Fig. 18B)

A principal component analysis based on the MD-trajectories of the Rab32:GtgE complex confirms that the Rab32:GtgE complex is preserved in the GDP-state. However, replacement of the nucleotide for GTP alters the complex behavior and reveals loop-motions of Rab32 and GtgE into opposite directions (Supplementary Fig. 17). Consequently, the analysis suggests that Rab32 is binding to GtgE in the GDP-state but dissociates from the protease in the GTP-state.

In our *ca.* 200 ns MD-simulations, the global structure and dynamics of both Rab32 and GtgE remain similar, but we observe

an increase in distance between F88$_R$ and K194$_G$ in the GTP-bound complex (Fig. 5e, bottom and Supplementary Fig. 18A). The MD-simulations reveal that a third putative hub (hub 3) in the complex interface between switch II and spike 3 could be involved in the Rab32–GtgE recognition, comprising residues F88$_R$, K194$_G$, and E114$_G$ (Fig. 5b). In the MD-simulations with GTP, we observe an increase in the F88$_R$–K194$_G$ distance, which could lead to a weaker Rab32–GtgE interaction, consistent with the observed lack of Rab32:GTP-binding to GtgE in the in vitro experiments (Supplementary Fig. 18B). In order to probe the

function of $F88_R$ in Rab32, we replaced this residue with a glycine ($F88G_R$), and re-initiated the MD-simulations. Interestingly, the MD-simulation of the $F88G_R$ demonstrate that the GTP- and GDP-states behaves more similar in this mutant (Supplementary Fig. 18C). Therefore, to validate the significance of hub 3 for the GtgE selectivity towards GDP over GTP, we produced the Rab32 $F88G_R$-mutant. To this end, we purified the full-length Rab32 wild type and the $F88G_R$-mutant with GDP or GppNHp, challenged it with cleared *E. coli* cell lysates containing over-expressed full-length wild-type or GtgE-mutants ($E114A_G$, $K194A_G$, $E114A_G/K194A_G$), and analyzed Rab32-proteolysis using SDS-PAGE based gel shift (Fig. 5f and Supplementary Fig. 15B). Interestingly, although Rab32 $F88G_R$ generally becomes a worse substrate for GtgE, the $F88G_R$ mutation indeed results in a loss of the strong GDP-state preference of GtgE for Rab32 (Fig. 5f). The decrease in GtgE-efficiency for Rab32 $F88G_R$ is expected due to the involvement of $F88_R$ in the complex interface. However, the $F88G_R$-mutant is now cleaved both in the GDP- and GTP-states in contrast to the wild-type GTPase (Fig. 5f). Thus, the $F88_R$-moiety is obviously important in the switch II rearrangement during complex formation, as indicated by the MD-data.

In contrast, the alanine substitutions of GtgE $E114_G$ and $R194_G$ involved in hub 3 show opposite effects and are neutralizing each other in the double mutant when combined with Rab32 $F88G_R$ (Supplementary Fig. 15B, right). The increased space in the $F88G_R$ mutant may be required to form a catalytically competent Rab32:GTP:GtgE complex. We speculate that $E114A_G$ decreases the proteolytic activity emphasizing its vital role as electrostatic anchor for $K194_G$. When not fixed by the carboxylate function of $E114_G$, the ε-amino group of $K194_G$ is free to establish other intramolecular salt bridges or polar contacts with the neighboring $D193_G$ or $Y195_G$, and could thereby disturb the organization of the hub 3 interface. The Rab32 switch II interaction with spike 3 may thus be a crucial part in the nucleotide selection.

In conclusion, we provide insights into a multifactorial selection mechanism since no single tested hub was able to abolish the nucleotide substrate preference of GtgE entirely. Nevertheless, the computationally derived and experimentally verified $F88G_R$-mutation results in the proteolysis of both active and inactive Rab32.

## Discussion

In this work, we have biochemically and structurally characterized the Rab32:GDP:GtgE complex. We have established that GtgE specifically acts on inactive, GDP-bound Rab32.

Furthermore, based on combined atomistic molecular simulations and in vitro experiments, we could demonstrate that the $F88_R$ position in the switch II regulatory region of Rab32 contributes to a multifactorial activity-state discrimination mechanism by GtgE and that it forms a basis for the GDP-state specificity of the protease.

Only two other proteins are known to have exclusive affinity for inactive-state Rabs. Here, the functionally and structurally related regulatory factors Rab escort protein (REP) and GDI are involved in the prenylation and membrane delivery or membrane cycling of Rabs, respectively, and interact with low nanomolar affinity with Rab:GDP[24–26]. In particular, the physiological role of GDI is to effectively solubilize lipidated Rabs from intracellular membranes after the protein has returned to the inactive form. Even though the complexes of Rab-proteins with REP and GDI have been extensively investigated on a structural level, the molecular basis for the exclusive interaction with inactive Rabs remains elusive[20,27,28]. Our structural and molecular dynamics analysis of the Rab32–GtgE interactions, however, revealed that the side chain of $F88_R$ in Rab32 forms an important structural determinant for discriminating the active and inactive Rab GTPase.

It is interesting to note that the Rab-recruitment to intracellular membranes is dependent on membrane-localized GEFs. Therefore, Rab-targeting and activation are directly coupled[29]. Consequently, GEF-mediated Rab32 recruitment to the SCV would result in the creation of active Rab32:GTP that is not a GtgE-substrate and consequently would not be subject to proteolysis. Interestingly, Salmonella secretes another protein that ensures inactivation of Rab32:GTP, making Rab32:GDP accessible to GtgE. During infection, the Salmonella Rab32-GAP SopD2 is released and catalyzes the conversion of Rab32:GTP to Rab32:GDP[15]. Apparently, Salmonella causes effective GtgE-mediated Rab32-cleavage by ensuring the prior conversion of active into inactive Rab32.

The question arises as to why Salmonella has evolved with a GDP-state specific Rab-protease making it dependent on a second secreted protein (i.e. SopD2). We hypothesize, however, that the presence of SopD2 ensures time-efficient access to GDP-loaded Rab32 since the latter would be blocked by tightly binding to GDI. We therefore suggest the following order of events: The Rab-protein is targeted to and activated at the membrane by a corresponding GEF[29]. This successive pathway of GEF-mediated Rab32 activation/recruitment and SopD2-mediated Rab32-inactivation results in direct access of GtgE to Rab32 (Fig. 6). In the hypothetical case where GtgE was to attack Rab32:GDP without prior action of GEFs and SopD2, it would need to directly

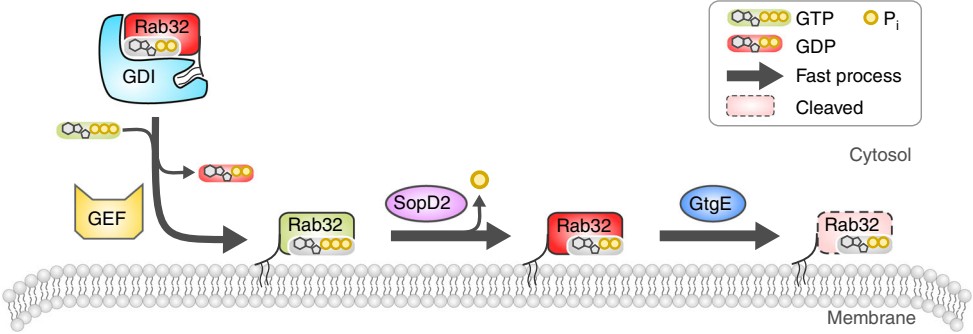

**Fig. 6** Mechanistic model of the dependence of GtgE-mediated Rab32-cleavage on SopD2. The cooperation of SopD2 and GtgE allows rapid Rab32-inactivation since the proteolysis is a result of a series of fast enzymatic conversions. Rab32 is recruited to the membrane by a corresponding GEF (BLOC-3). The resulting SCV-bound GTP-loaded GTPase is then rapidly converted by the action of the GAP SopD2. Eventually, Rab32:GDP is cleaved quickly by GtgE. GEF guanine-nucleotide exchange factor, GDI GDP-dissociation inhibitor

compete with GDI for Rab32-binding in the cytosol. Since GDI and GtgE occupy the identical binding site on Rab32 and since the interaction of Rabs with GDI are of high nanomolar affinity, GDI would effectively shield Rab32 from GtgE[20,26,27]. The high affinity of the Rab32:GDP–GtgE interaction (indicated by a nanomolar $K_D$, $K_{D, Rab32:GDP:GtgE} = 100$ nM) would not be sufficient to displace GDI ($K_{D,Rab:GDI} < 5$ nM) in the cytosol and would possibly not allow time-efficient Rab32-cleavage.

In contrast, the combination of SopD2 and GtgE Salmonella at the SCV membrane during infection guarantees fast Rab32-inactivation as a results of a series of rapid enzymatic steps (Fig. 6): Rab32:GDP is likely recruited to the SCV via a GEF (BLOC-3) from its cytosolic complex with GDI promptly. The resulting pool of active Rab32:GTP is then immediately targeted by the GAP SopD2, leading to GTP-hydrolysis and the production of membrane-localized Rab32:GDP. Finally, GtgE is attacking Rab32:GDP at the SCV and the proteolyzed GTPase is formed quickly. Therefore, the cooperation of SopD2 and GtgE may overall be the most efficient way for quantitative Rab32-proteolysis (Fig. 6). An efficient Rab-inactivation therefore appears to be crucial for the progressing SCV maturation.

Together with previous findings, our present results contribute to the understanding of the order of molecular events at the SCV: the maturation of the SCV is dependent on the small GTPases Rab7 and Rab9[30–32]. After activation, Rab9 may recruit the Rab32-GEF BLOC-3 to the surface of the SCV[33,34]. BLOC-3 subsequently recruits Rab32 to the SCV from its complex with GDI and activates the G-protein by loading with GTP. In order to stably establish the infection, Salmonella counteracts Rab32-dependent membrane pathways with SopD2 and GtgE, leading to Rab32 conversion into the GDP-state and irreversible proteolysis in the switch I region, respectively[5,15].

It is interesting to note that despite of Rab32 recruitment to the SCV, the protein is not stably accumulated at the membrane in Salmonella Typhimurium (containing the GtgE and SopD2 proteins), indicating that GtgE-cleaved Rab32:GDP is lost from the vacuole in an active or passive process[5]. Likely, the Rab32-proteolysis is not affecting intrinsic membrane affinity since the C-terminal lipids remain unaffected, making a direct GtgE-mediated Rab membrane release improbable. Since our results demonstrate that cleaved Rab32 remains a stable protein that alsocan be purified in vitro, active processes to remove the Rab-protein from membrane may be ongoing. Here, ubiquitination in conjunction with proteasomal degradation or membrane extraction by GDI are conceivable mechanisms. Cleaved Rab32 may still interact with and be extracted by GDI, leading to effective removal of the GTPase. GDI is only weakly interacting with the switch I regions of Rab-proteins and therefore presumably tolerating the proteolytic modification[20].

We speculate that once extracted by GDI, the GtgE-cleaved Rab32 is presumably not recruited by BLOC-3 to the SCV again. Although the 3D-structure of BLOC-3 is unknown, its sequence similarity to components of the Ypt7–GEF-complex Mon1/Ccz1 suggests that the interaction with Rab32 happens in a mechanistically identical manner[35]. In contrast to the Rab–GDI interactions, the switch I loop intimately interacts with the GEF. Consequently, Rab32-proteolysis is potentially incompatible with activation by BLOC-3, and thus re-recruitment of GDI-bound Rab32 would be prohibited.

In conclusion, GtgE is a GDP-state specific Rab32-protease having a high affinity for its substrate. Structural studies and MD-simulations identify an amino acid hub that contributes to the selective binding of the inactive state of Rab32. The analysis provides insights into the order of protein interactions involved at the SCV and imply a requirement for the Salmonella GAP SopD2 in order to permit GtgE-mediated Rab32-proteolysis. It will be interesting to investigate how the GtgE-processed Rab32 contributes further to the infection process.

## Methods

**DNA cloning.** For all cloning tasks, either XL1 blue or Mach1 E. coli strains were used. Site and ligation independent cloning (SLIC) approach was applied for all molecular cloning. The gtgE gene from Salmonella enterica serovar Typhimurium str. SL1344 (CBW17091.1) was synthesized with an optimized sequence for E. coli codon usage provided in a standard cloning vector (Life Technologies). The sopD2 gene (WP_001145561.1.) was amplified from genomic DNA from the same strain. Both Salmonella genes were subcloned for heterologous expression into a modified pMAL-c2X vector (Addgene ID 75286) harboring an N-terminal His6-MBP tag (MBP: maltose binding protein) fused through a TEV protease cleavage site between the 3′-NdeI and 5′-HindIII restriction sites. Additionally, truncated Rab32$_{18-201}$ and GtgE$_{21-214,C45A}$ were generated in the pMAL vector as described above and used for protein crystallization and structural studies, only. For Yeast-Two-Hybrid experiments, gtgE was introduced to the pGAD-HA vector (Clontech, Cat. No 630442) harboring an N-terminal activation domain. The human Rab GTPases (Rab32, NP_006825.1; Rab29 also Rab7L1) NP_001129134.1; Rab38 NP_071732.1; and Rab23 NP_057361.3) were introduced into the pMAL vector with 3′-NdeI and 5′-XhoI restriction sites. Fusion constructs of the pLexA vector (Addgene ID 11342) with all mentioned GTPases harbored an N-terminal DNA binding domain lacking the C-terminal prenylation site cysteines. Mutations were introduced with the Q5® Site-Directed Mutagenesis Kit (NEB) following the manufacturer's procedure. All plasmids were verified by DNA sequencing (GATC Biotech).

**Recombinant protein expression and purification.** All proteins were recombinantly expressed in E. coli BL21(DE3)-RIL chemically competent cells (Agilent) growing in LB broth supplemented with ampicillin. The expression culture was inoculated with 2–5% (v/v) from a pre-culture and grown at 37 °C shaking at 180 r. p.m.. Expression was induced at an OD$_{600}$ of 0.5 by addition of 0.3 mM IPTG (final). After induction the culture rested at 4 °C for 30 min and was grown 16–20 h at 22 °C and collected at 5900 × g for 25 min. The pellet was washed with phosphate-buffered saline (1xPBS) and collected again under same conditions. Cell pellets were snap frozen in liquid nitrogen and stored at −20 °C until further use. Pellets of GtgE and SopD2 proteins were resuspended in 10 mL Buffer A (50 mM Hepes pH 7.5, 500 mM NaCl, 2 mM β-mercaptoethanol (BME)) for each gram cells and lysed mechanically with one passage through a fluidizer at 2.0 kbar (Constant systems). Lysate was cleared by centrifugation 48,000 × g for 30 min and the supernatant was applied to a 5 mL Ni$^{2+}$-immobilized metal-affinity chromatography column (IMAC) column (HiTrap, GE Healthcare) previously equilibrated with Buffer A. The column was washed with 6 column volumes (CV) 5% Buffer B (Buffer A supplemented with 500 mM imidazole) and protein of interest was eluted applying a gradient up to 50% Buffer B in 10 CV. Fractions containing the His-MBP tagged protein of interest were pooled and dialyzed over night at 4 °C against Buffer C (20 mM Hepes pH 7.5, 50 mM NaCl, 2 mM BME) and subjected to Tobacco Etch Virus (TEV)-protease cleavage. Cleaved protein mixture was applied a second time on an IMAC column to remove the TEV protease and His-MBP tag. The highly pure protein was collected in the flow through, concentrated, snap frozen and stored at −80 °C.

Purification of Rab GTPases: The procedure was the same as described for the GtgE purification above except all buffers were supplemented with 1 mM MgCl$_2$ and 10 μM GDP and after cell lysis 1 mM phenylmethylsulfonyl fluoride (PMSF; final) was added before centrifugation. To reach homogeneity after the IMAC Rab GTPases were further subjected to size-exclusion chromatography (SEC) on a Superdex 75 (16/600) (GE Healthcare) pre-equilibrated with SEC-buffer (20 mM HEPES, pH 7.5, 50 mM NaCl, 1 mM MgCl$_2$,10 μM GDP, and 1 mM DTT). Fractions containing monomer were concentrated, snap frozen and stored at −80 °C. The identity of all proteins was verified by mass spectrometry using an electron-spray-ionization mass spectrometer coupled to a liquid chromatography (LC-ESI-MS; LCQ fleet; Thermo Scientific).

For the in vitro complex formation, 8 μM Rab GTPase was incubated with 8 μM GtgE$_{C45A}$ in SEC-buffer for 30 min at room temperature. For analysis 50 μL of the mixture was subjected to an analytical gel filtration (Superdex 75 13/30, GE Healthcare) pre-equilibrated with SEC-buffer with a flow of 0.5 mL/min recording the absorption at 280 nm. As negative controls, the respective monomers were analyzed identically.

**Yeast-Two-hybrid interaction studies.** For the Y2H assay, pLexA-GTPase plasmids (bait) encoding for full-length Rab proteins were transformed with MAT-a yeast strain Y187 adopted from a previous method[36] and plated on selection dropout medium lacking tryptophan (SD-W)[37]. In brief, 700 μL of stationary yeast culture was collected at 375 × g for 2 min and vortexed with 100 μL sterile one-step buffer (0.2 M lithium acetate, 40% PEG 3350, and 100 mM DTT). Subsequently 100–500 ng of respective plasmid DNA was added and incubated at room temperature for 15 min before heat shock was applied for 30 min at 45 °C. The pGAD-gtgE$_{C45A}$ plasmids (prey) encoding for inactive full-length GtgE protein were transformed the same way with MAT-α S. cerevisiae L40ΔGal4 and were grown on

SD-L plates (lacking leucine) at 30 °C for 3–4 days. For yeast mating, single colonies of a prey and a bait clone were grown separately in appropriate selection medium for 2 days while shaking. To 100 μL YAPD media[37] in a 96-well plate (Sarstedt) 50 μL of bait and prey culture were added sequentially and incubated 22–24 h at 30 °C with 180 r.p.m. Cells were resuspended and 5 μL were spotted on SD-LW plates as mating control and SD-LWH (lacking histidine) for phenotypic read out. Plates were analyzed after 3–4 days of incubation at 30 °C. 3-Aminotriazol (3′-AT, Sigma Aldrich), an inhibitor of the histidine biosynthesis, was added to SD-LWH agar plates in the range of 0–10 mM probing the interaction intensity qualitatively.

**Isothermal titration calorimetry.** Interaction studies by isothermal titration calorimetry (ITC) were conducted on a ITC200 microcalorimeter (MicroCal). Measurements were performed in SEC-buffer at 37 °C. Rab32$_{FL}$:GDP was diluted to 20 μM and GtgE$_{FL,C45A}$ was introduced to the injection unite with 200 μM. Injection volume was set to 1.5 μL per injection and the heat power was recorded over time until binding saturation was obtained. The binding isotherms were integrated, corrected for the offset, and the data were fitted to a one-site-binding model using the ITC analysis software provided by the manufacturer (MicroCal), yielding the equilibrium-dissociation constant ($K_D$).

**Analytical ultracentrifugation.** Interaction studies by analytical ultracentrifugation (AUC) were conducted in a Beckman ProteomeLab XL-A analytical ultracentrifuge (Beckman Coulter) equipped with a fluorescence detection system (Aviv AU-FDS; Aviv Biomedical) as described previously[38]. A constant concentration of 200 nM Rab32$_{FL}$-Atto488 in SEC-buffer was incubated with varying molar ratios of GtgE$_{FL,C45A}$ as indicated for 2 h in the dark at room temperature, ensuring equilibrium conditions prior to measurement. The data were analyzed with SedFit software[39]. The sedimentation coefficients and molecular weights of the present species were obtained by solving the Lamm equation. The fraction of labeled Rab-proteins bound to GtgE$_{C45A}$ was plotted against the GtgE$_{C45A}$ concentration and data were fitted with a hyperbolic function for calculation of the $K_D$.

**Nucleotide loading of GTPases.** Nucleotide exchange of small GTPases with GDP or GTP (Carbosynth) was performed in SEC-buffer supplemented with 10 mM EDTA and a 40-fold molar excess of nucleotide over protein and incubation at 15 °C for 5–16 h before performing size-exclusion chromatography (Superdex 75 16/60; GE Healthcare). Nucleotide exchange with the non-hydrolyzable analog GppNHp (Jena Bioscience) was performed by addition of equimolar amounts to the GTPase with 5 U Antarctic Phosphatase (NEB) per mg protein in nucleotide exchange buffer (50 mM Tris HCl pH 8.0; 200 mM (NH$_4$)$_2$SO$_4$; 10 μM ZnCl$_2$). Accidentally precipitated protein was removed by centrifugation and a buffer exchange was performed on a NAP 5 desalting column (GE Healthcare) according to the manufacturer's instructions with SEC-buffer containing 1 μM GppNHp. Protein containing fractions were pooled, concentrated, snap frozen, and stored at −80 °C. Nucleotide loading efficiency was tested by ion-pairing reversed-phase high performance chromatography (RP-HPLC; Prontosil C18, Bischhoff Chromatography) in 50 mM potassium phosphate buffer pH 6.6; 10 mM tetra-butylammonium bromide and 12% acetonitrile (v/v). Protein samples (25 μM, 30 μL) were heat precipitated at 95 °C for 5 min and cleared by centrifugation for 10 min at 48,000 × g. Supernatant was subjected to chromatographic separation and nucleotide peaks were integrated and normalized to the total amount of nucleotides detected set to 100%. Nucleotide retention times were determined with the respective nucleotide standard in a separate run.

**Kinetics of Rab32 cleavage.** Cleavage of Rab GTPases by GtgE was monitored by SDS-PAGE gel shift assay. Typically 8 μM of Rab$_{FL}$ was cleaved by 8 nM GtgE$_{FL}$ in SEC-buffer supplemented with 10 μM GDP, 10 μM GTP or 1 μM GppNHp respectively at 25 °C. At different time points, 10 μL samples (ca. 2 μg) were taken an the reaction was quenched immediately with pre-heated Laemmli buffer and 3 min boiling at 95 °C. For quantification purpose 2 μM, MBP was added as internal standard into the reaction mixture prior to start of the experiment. After Coomassie staining (Brilliant blue R-250, Thermo Scientific), band intensities were determined by densitometry from gray values with image studio lite software (LI-COR). Rab-bands were normalized with the respective MBP-band intensity and plotted over time. The apparent rate constant ($k_{obs}$) was obtained by fitting the data exponentially and $k_{cat}/K_M$ resulted from dividing $k_{obs}$ by GtgE concentration [$E_0$]. For rapid qualitative analysis of GtgE-mutant activity, the method was adopted as follows: GtgE variants were recombinantly expressed in 50 mL. Cells equal to 5 OD units were resuspended in 0.5 mL SEC-buffer lacking nucleotides and lysed by ultrasonication (Digital Sonifier® S-450, Branson Ultrasonics). Protein expression levels were monitored from the cleared lysate by SDS-PAGE and total protein amounts were determined with the OD at 280 nm. Subsequently, the lysate was introduced directly to the assay as described above (ca. 2 μg in 10 μL reaction) or diluted 1:100 in nucleotide free SEC-buffer prior to use.

**GtgE-mediated Rab32 cleavage.** The applicability of fluorescence-based methods for monitoring GtgE-mediated Rab32-proteolysis was evaluated using a potential change in intrinsic Rab32-tryptophane fluorescence or a change of mant-fluorescence for Rab32 loaded preparatively with the fluorescent GDP-derivative mantGDP.

Cleavage experiments of Rab32:GDP by GtgE were performed at 25 °C in a 1 mL quartz cuvette using a fluorescence spectroscope (FluoroMax®-4, Horiba Ltd., Japan). All measurements were performed in filtered degassed assay buffer (20 mM HEPES, pH 7.5; 50 mM NaCl; 1 mM MgCl$_2$; 1 mM DTT) with nucleotides added right before the measurement. For tryptophane fluorescence experiments, 500 nM Rab32:GDP was added to the assay buffer supplemented with 10 μM GDP. Samples were excited at 297 nm (1 nm band-pass) and emission was recorded at 340 nm (5 nm band-pass). For mant-fluorescence experiments, 500 nM Rab32:mGDP was added to the assay buffer supplemented with 20 nM mGDP. Samples were excited at 340 nm (1 nm band-pass) and emission was recorded at 440 nm (5 nm band-pass). GtgE was added at 10 nM as final concentration to start the reaction (GtgE: Rab molar ratio 1:50).

**Covalent protein fluorescence labeling.** GtgE$_{FL,C45A}$ was buffer exchanged to labeling buffer (20 mM HEPES pH 7.5, 50 mM NaCl, 1 mM MgCl$_2$; 1 mM Tris(2-carboxyethyl)phosphine (TCEP)) and subsequently labeled covalently with the primary lysine amines using NT-647-NHS labeling kit (Nanotemper) following the manufacturer's instructions. The reaction was quenched by addition of 10 mM Tris HCl pH 7.5 and buffer exchanged on a desalting column for labeling buffer. Rab32$_{FL}$ was exchanged to labeling buffer supplemented with 10 μM GDP. Thiol groups in Rab32$_{FL}$ were covalently labeled with 1.1 molar excess of Atto488-Maleimide (ATTO-TEC) incubating at 4 °C for 16 h and the reaction was quenched by addition of 0.5 mM DTT.

**Lysine methylation of the Rab32:GDP:GtgE complex.** Methodology was adopted from a previous publication[16]. The lysine methylation reaction was performed in SEC-buffer with the preformed full-length Rab32:GDP:GtgE$_{C45A}$ and Rab32$_{18-201}$: GDP:GtgE$_{21-214,C45A}$ hetero complexes (1:1 molar ratio) and 2 mg mL$^{-1}$ total concentration. 20 μL freshly prepared 1 M dimethylamine-borane complex in reaction buffer (5 μL final; ABC; Sigma Aldrich) and 40 μl 1 M formaldehyde (8 μL final; made from 37% stock in reaction buffer; Sigma) were added per milliliter protein solution, and the reaction was gently rotated at 4 °C for 2 h. A second addition of ABC and formaldehyde followed a 2 h incubation under same conditions. Following a final addition of ABC the reaction was rotated overnight at 4 °C. Precipitations were removed by centrifugation and the supernatant was applied to a gel filtration (Superdex 75 16/60, GE Healthcare) pre-equilibrated with SEC-buffer. Fractions containing the hetero-complex were concentrated to a total of 10 mg/mL and immediately applied to protein crystallization.

**Differential scanning fluorimetry.** For the protein stability measurements, differential scanning fluorimetry (DSF) was conducted on a real time PCR device (Mx3005P, Agilent). In each well of technical replicas, 2 μg of Rab32$_{FL}$:GDP (full length or cleaved) was prepared in 20 μL SEC-buffer supplemented with 1-fold SYPRO Orange (Sigma Aldrich, 5000-fold stock). The mixture was heated in 1 °C min$^{-1}$ steps from 25–95 °C while recording the fluorescence signal with the excitation at 465 nm and the emission at 590 nm. The protein melting temperature ($T_m$) was obtained by fitting the data with the Boltzmann fit.

**Structure determination by X-ray crystallography.** Crystallization trials of full-length and truncated human Rab32:GDP (residues 18–201 in the truncated construct) complexed with the full-length and truncated protease GtgE$_{C45A}$ mutant from *Salmonella enterica* ssp. Typhimurium (residues 21–214 in the truncated complex) were performed by the sitting-drop vapor diffusion method, using protein concentrations of 10 mg/mL. Initial sparse–matrix screens with 0.2 μL of protein and 0.2 μL of reservoir solution identified several hits, but these crystals did not diffract X-rays beyond 5 Å resolution. Reductive methylation on surface exposed lysine residues significantly improved the diffraction quality of the full length and truncated Rab32:GDP:GtgE-complexes[16]. Suitable crystals of the full-length complex were grown in 0.05 M imidazole pH 8, 20% PEG 6000; crystals of the truncated complex were obtained from 0.1 M Bis-Tris pH 5.5, 25% PEG 3350. Crystals of both complexes were cryoprotected in reservoir solution supplemented with 25% ethylene glycol and flash cooled in a stream of liquid nitrogen gas at 100 K.

**Diffraction data collection and structure determination.** The data sets of full length and truncated Rab32:GDP:GtgE complexes were collected from single crystals using synchrotron radiation ($\lambda = 1.0$ Å) at the X06SA-beamline (Swiss Light Source, Villingen, Switzerland). X-ray intensities were assessed and data reduction carried out with XDS and XSCALE[40]. Crystals of the full-length complex diffracted to a resolution of 2.9 Å (space group $P6_5$22 with $a = b = 67.4$ Å, $c = 427.3$ Å); crystals of the truncated complex diffracted to a resolution of 2.3 Å (space group $P2_12_12_1$ with $a = 47.8$ Å, $b = 66.9$ Å, $c = 110.9$ Å).

Initial phases were calculated for the Rab32$_{18-201}$:GDP:GtgE$_{21-214,C45A}$ complex by Patterson search procedure using the deposited coordinates of Rab32:GppCH$_2$p (residues 18–201; PDB ID: 4CYM) and a partial model of GtgE (residues 80–213; PDB ID: 4MI7). The starting models were sequentially placed as a heterodimer into the asymmetric unit using PHASER (TFZ score = 11.7)[41]. After model building

with COOT[42], rigid body refinement and preliminary positional refinement with phenix.refine[43], the GtgE subunit was re-traced. Application of feature-enhanced maps using PHENIX in the early and intermediate stages of model building and refinement aided greatly in extending the model to finally include residues 23–213 of GtgE and residues 22–195 of Rab32, along with the GDP and magnesium ion co-factors of the latter protein[44]. Model refinement was performed in phenix.refine and REFMAC[45]. Eventually, water molecules were placed automatically using ARP/wARP[46]. The final model of the Rab32$_{18–201}$:GDP:GtgE$_{21–214,C45A}$ complex, with refined Translation/Libration/Screw (TLS) tensors and individual B-factors, converged to $R_{work}$ and $R_{free}$ values of 20.8% and 24.5%, respectively. Coordinate geometries were confirmed via MOLPROBITY to possess good stereochemistry and small bond-length and angle RMSDs, with a single glycine residue 59 of Rab32 —adjacent to the site of proteolytic cleavage by GtgE—in the disallowed region of the Ramachandran plot[47]. Magnesium ion coordination was verified with the CheckMyMetal server[48].

The full-length Rab32:GDP:GtgE$_{C45A}$ structure was solved by molecular replacement, applying the refined coordinates of the Rab32$_{18–201}$:GDP:GtgE$_{21–214,}$ $_{C45A}$ heterodimer as the initial search model. A single copy of the complex constituted the asymmetric unit of the crystals belonging to space group $P6_522$. In this case, model building was straight forward, eventually featuring residues 23–217 of GtgE and residues 21–195 of Rab32. TLS refinement yielded converging values for $R_{work} = 20.1\%$ and $R_{free} = 24.0\%$ along with low bond-length and angle RMSDs. Inspection of the Ramachandran plot showed good stereochemistry, with glycine residue 59 of Rab32 again the sole outlier. Structural representations of proteins were prepared with PyMol[49] or Visual Molecular Dynamics (VMD) software[50].

**Molecular dynamics simulations.** MD-simulations were performed based on our resolved X-ray structure of the truncated Rab32:GDP:GtgE$_{C45A}$-complex (i.e., Rab32$_{18–201}$:GDP:GtgE$_{21–214,C45A}$) by replacing Ala-45 with Cys in order to model the wild-type GtgE. The protein complex was embedded in a water-ion environment with a 150 mM NaCl concentration, and the complete simulation setup comprised 98,980 atoms. The wild type and F88$_G$ truncated Rab32:GtgE complexes were simulated in presence of both GTP and GDP for 190 ns at $T = 310$ K using a 1 fs time step, by using the CHARMM36 force field[51], and treating the long-range electrostatics using the Particle Mesh Ewald (PME) approach[52]. The MD-simulations were performed using NAMD[53], trajectories were analyzed using VMD[50], and principal component analysis based on the MD-trajectories were performed using ProDy[54,55].

**Data availability.** Structure factors and model coordinates for the full length and truncated Rab32:GDP:GtgE$_{C45A}$-complexes have been deposited in the RCSB Protein Data Bank under the accession codes 5OED and 5OEC, respectively. The data that support the findings of this study are available from the corresponding author upon request.

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

## Acknowledgements

This work was performed in the framework of SFB 1035 (German Research Foundation DFG, Sonderforschungsbereich 1035, projects A02, B05, and B12). Computational resources for this project were provided by the Gauss Centre for Supercomputing/Leibniz Supercomputing Centre (grant: pr84pa). We thank the staff of the beamline X06SA at the Paul Scherrer Institute, Swiss Light Source for their assistance in X-ray data collection. Dan Humphreys is acknowledged for critically commenting on the manuscript. Christine John and Maximilian Biebl are acknowledged for analytical ultracentrifugation experiments.

## Author contributions

A.I.: Conceived research. R.W., B.B., S.M., F.E., V.K., M.G., A.I.: Designed the experiments, analyzed and interpreted the data, and wrote the manuscript; R.W., B.B., S.M. and F.E.: Performed experiments.

## Additional information

**Competing interests:** The authors declare no competing financial interests.

