## [Peer Review File · Nature Communications]

Reviewers' comments:

Reviewer #1 (Remarks to the Author):

This manuscript presents interesting structural and functional data on the enzyme GtgE, which is secreted by Salmonella and targets a subset of Rab GTPases. The structure of the enzyme (unbound) and its specificity had already been determined - the contribution of this report is the detailed analyses of the complex between the enzyme and substrate, Rab32(GDP), the affinity and kinetic properties of the enzyme, and the molecular basis for substrate and GDP specificity. GtgE not only prefers a subset of Rabs, but it specifically targets the GDP form of the Rabs. The structural, enzymatic and biochemical assays, in concert with molecular dynamics simulations, provide a model for these features of the enzyme. However, there are several shortcomings that need to be addressed, and the findings also require a better connection to the biological context of the Salmonella life cycle.

The enzymatic characterization was challenging, given the lack of a simple spectrophotometric assay. The nearly 7-fold difference between AUC and MST is, however, quite significant and perhaps should not be glossed over in the paper. The AUC measurement and ITC are much closer in agreement (0.1 vs. 0.35 micro molar). The fluorescent dye NST647 is used for MST, and could be significantly decreasing affinities by the modification of lysine residues. Perhaps the authors could perform an ITC titration with NST647-modified Rab, and see whether the affinity drops? The large difference leads to significant spread in the affinity and kinetic parameters, which has an impact on how to reconcile the differences in affinities between SopD2 and GtgE. Essentially, my concern is that MST is not providing a reliable affinity, relative to the other methods.

Also, the ITC data appears to be from a single measurement, and without a control (GtgE into buffer). A control seems warranted in this case, as there appear to be heat effects from dilution of ligand at the end of the titration. The errors shown (Fig 2C) seem to be from the fit to data of a single set of titrations. In this respect, ITC data is an 'outlier' relative to MST and AUC in not having been repeated for estimates of random error (rather than presumably systematic errors coming from plot fitting shown). I think it would be useful to resolve these issues with some control experiments and repetitions to narrow the spread of kinetic parameters.

Placing narrower spread for affinity is also important in understanding how Gtg could access Rab32(GDP) as it must compete with SopD, which has nM affinity for the Rab (Discussion).

Also - why is it an evolutionary disadvantage to have a 'second secreted protein'? Is it simply the extra genetic load and the trouble of protein synthesis/secretion?

In addition, can the authors comment on how Gtg is able to cleave Rab32 when the affinity is at least 50-fold weaker, relative to GDI? Are there possibly separate pools of inactive Rab32 in infected cells? The authors vaguely hint at a 'cooperation' of GtgE and SopD2, but this should be clarified further. It is suggested in the discussion that SopD2 could extract GtgE-cleaved Rab32, since the SopD2-Rab32 interaction is not significantly dependent on switch I. Is there experimental evidence that the GDI can interact with cleaved Rab32(GDP)? The suggestion that cleaved Rab32 could be extracted by SopD2 from the SCV implies that GtgE can cleave membrane bound Rab32(GDP). Given the slow intrinsic GTPase activity of Rab32 (Fig 1C), how is the substrate for GtgE generated?

In summary, the findings from in vitro assays and the crystal structure need better connections to biology, i.e., how does GtgE contribute to Salmonella infections? As it stands, the manuscript certainly gives a better understanding of the in vitro catalytic activity of the enzyme. Although clearly interesting and novel, the main shortcoming of the paper is how that activity fits into the life cycle and host range of Salmonella infection.

The other main issue that I have with the manuscript concerns the MD work. The authors start with the pre-formed complex, but as comparison, substitute GTP for GDP to evaluate specificity. However, Rab32(GTP) does NOT form a complex with GtgE, therefore its complex with GtgE (at start of simulation) is artificial. Furthermore, Rab32(GTP) conformation is distinct from Rab32(GDP), and so this experimental strategy seems highly contrived. The simulation does appear to succeed in identifying a determinant of GTP specificity, and is subsequently supported by mutagenesis and enzymatic assays. However, this positive result does NOT obscure the fact that the MD strategy is poorly designed. The current simulation may show why GTP destabilizes the Rab32 (starting from a GDP state), but is NOT designed to provide insight into pre-binding preferences for GDP vs. GTP conformations. The key question of pre-binding compatibility (GDP vs. GTP) cannot be addressed with this kind of simulation.

I can outline several suggestions:

- at the very least, the authors must state the shortcomings in this simulation and explain clearly why they chose this strategy
- the structure of Rab32(GTP) is known. An alternative simulation is to 'steer' the complex towards the GTP conformation (but keep the GDP in complex), and see how the system reacts. Still not ideal, but there is less chance for the simulation 'exploding'
- detach the complex a small distance, and steer either Rab32(GDP) or Rab32(GTP) toward the known structure of the complex. This would, in principle, provide the best simulation of the binding preferences. However, this would require major investment in computing time and is

more 'artificial' and likely to fail.

- it might be useful to validate simulations by looking at the binding affinity of the F88G mutation of Rab32 in the GTP state. Comparisons of GTP-bound F88G would provide significant support for the hub3/spike contribution to specificity. Presumably it would be a weak interaction, but certainly more than wild-type Rab32(GTP). Clearly the mutant is enabling cleavage of the GTP conformation (Fig 5F), therefore binding affinity is significant and can be quantified. The experiment would provide considerable support for the model for GDP selectivity.

A few other comments and suggestions:

- quality and coloring of molecules (ribbons, Fig 3) need revisions, difficult to tell Rab/GtgE apart at the interface. The interior surface of Gtg ribbon is the same color as Rab32 (grey), so the colors run into each other. Similar problems with 1B, 1C (superpositions) - ribbons should be a single color (both surfaces), and color choice should contrast better to identify molecules.

- similar objections to Fig 5B, very difficult to see superpositions, the ribbons (interior/exterior) must be uniform color to reduce complexity

-

grammar - rephrase 'would result in the creation of active Rab32:GTP that is no GtgE-substrate' to

'would result in the creation of active Rab32:GTP that is not a GtgE-substrate'

typos and technical issues:

- spelling 'Rab32-clevege' to 'Rab32-cleavage', in 'GtgE has high affinity...' section

- typo - 'an hydrophobic => a hydrophobic cluster' during structural description of interface

- 'obvisouly the recruitment of Rab32 to the SCV'. Also, why is Rab32 recruitment to SCV 'obviously' important?

- the average B-factors are very high for the full-length GtgE complex, relative to other structures at 3A resolution. Can the authors comment on this? In particular, the nucleotide average is 114, it might be useful to show the density in this region since it is somewhat irregular to have such high B-factors

Reviewer #2 (Remarks to the Author):

The manuscript by Wachtel et al. details comprehensive biochemical and structural studies and complementary molecular dynamics simulations that mechanistically demonstrate how a critical bacterial virulence factor binds to and cleaves host Rab GTPases. The bacterial pathogen *Salmonella typhimurium* expresses two virulence factors a RabGAP (SopD2) and a cysteine protease known as GtgE. Together these factors serve to inactivate host Rab GTPases that support pathogen elimination through phagolysosomal fusion. In particular, the GtgE protease supports intracellular survival and replication through the cleavage of host Rab GTPases. GtgE is essential to the expanded host range of *Salmonella typhimurium* relative to *S. typhi*. Even while two prior structural studies of proteolytic fragments of GtgE identified it as a cysteine protease, details of Rab substrate interaction and catalytic specificity remained unclear due to the use of peptide substrates rather than intact Rab GTPase proteins.

The present determination of the mechanistic underpinnings of GtgE proteolysis through biochemical and structural analyses represents an important advance. The work clearly identifies GDP-bound Rab GTPases (Rab29, Rab32 and Rab38) as the substrates of full length GtgE. The work also clarifies the additive virulence benefit to *Salmonella* strains that express a RabGAP to increase the Rab-GDP substrate pool, and GtgE to promote Rab proteolysis. It is speculated that even though GtgE cleavage does not destabilize the Rab it prevents further GEF mediated activation of the cleaved GTPase. Additionally, the study also identifies a phenylalanine in the switch 2 domain of Rab32 as critical to activity-state discrimination, which may be generalizable to GDI protein selectivity for Rab-GDP.

Overall, the work is highly compelling, well controlled and statistically validated. Molecular dynamics predictions are used to augment mutational analyses and to explicitly test specific predictions. The findings are groundbreaking, considered of significant interest to the Nature Communications readership and therefore highly recommended pending clarification of the text regarding which experiments were performed with FL vs. mutant proteins and interpretations of the mutant analyses in the context of non-binding Rab23.

Specific Comments

Methods and results sections indicate that both FL and truncated proteins were purified and used for structure determinations. In the Results section and for each figure indicate clearly which proteins were used, e.g. Fig. 4 shows truncated protein structure, yet text refers to wild-type GtgE on line 315. Under DNA cloning (lines 564-579) include the methods for construction of the truncated proteins.

The work considers GtgE interactions with Rab32, Rab38 and Rab29 as contrasted to non-

binding Rab23. However, there is a missed opportunity in the analyses to explain the selectivity for binding to particular Rabs and which amino acids might strengthen affinity. For example, different mutants are used for yeast two hybrid studies to measure altered Rab32-GtgE binding relative to GtgE proteolysis, e.g. R87A and L64A mutants were used for binding and R87E and L64E were used for proteolysis studies. When comparing the sequence alignments of the 4 rabs in Fig. S1 it appears that there may be interesting differences relative to Rab23 in the beta2 and beta3 sequences that could account for the binding specificity of GtgE for Rab32, Rab38, and Rab29. As presented, because the alanine mutations don't block binding, one cannot surmise if the glutamate substitutions block binding and cleavage or just proteolytic activity. In line 325 it is stated that specific amino acids cause a growth defect confirming role in Rab32:GtgE complex formation. However, all of these amino acids are present in Rab23 so why doesn't it bind GtgE? Similarly line 338, W80 and F62 are conserved in Rab23. Perhaps there are some key amino acids in the beta2-beta3 regions that may play a role?
Line 348—need to more clearly distinguish the mutants that are involved in substrate recognition.

Minor Corrections

Introduction: line 93 change 'deactivation' to inactivation
Fig. 1 line 146 change 'means of a technical replica' to means of technical replicates
Line 161 Rab32-cleavage educts should be Rab32-cleavage products
Line 164 To this purposes should be For this purpose
Line 171 change to: ...would require Michaelis-Menten-kinetics to be conducted at
Line 188: change to: ..determined to be Kd
Line 505: change to that is not a GtgE-substrate
Line 536: change to: despite of Rab32
Line 544 and 545: be more clear that this is speculative
Line 567: change to synthesized
Line 598: change to subjected to Tobacco Etch
Line 612: instead of one equivalent state that 8 nM GtgE was used

Supplemental section

Change Varients to Variants in the A and B headings of Supplemental Figure S10

Line 94: Change S7 to S5

Response to referees

The protease GtgE from Salmonella exclusively targets inactive

Rab-proteins

We would like to thank the referees for their time and effort. We appreciate their constructive comments on our manuscript. Based on the referee comments we have optimized several aspects of our findings and have particularly improved the presentation of the data and the clarity of our arguments. We hope that the referees will concur with the changes to the manuscript.

Largely, the reviewers have raised three general points of criticism:

- a) Improving the interpretation of the molecular results and their significance in the context of Salmonella infections.
- b) Substantiating and specifying the affinity of the Rab32:GDP:GtgE complex.
- c) Criticism on the design of the molecular dynamics simulation of the hypothetical Rab32:GTP:GtgE complex.

We have addressed these comments below. In order to facilitate reviewing our response, we have applied the following color coding:

Reviewer comments: Black
Author responses: Blue
Text changes in the revised manuscript: Red

Reviewer #1 (Remarks to the Author):

This manuscript presents interesting structural and functional data on the enzyme GtgE, which is secreted by Salmonella and targets a subset of Rab GTPases. The structure of the enzyme (unbound) and its specificity had already been determined - the contribution of this report is the detailed analyses of the complex between the enzyme and substrate, Rab32(GDP), the affinity and kinetic properties of the enzyme, and the molecular basis for substrate and GDP specificity. GtgE not only prefers a subset of Rabs, but it specifically targets the GDP form of the Rabs. The structural, enzymatic and biochemical assays, in concert with molecular dynamics simulations, provide a model for these features of the enzyme. However, there are several shortcomings that need to be addressed, and the findings also require a better connection to the biological context of the Salmonella life cycle.

The enzymatic characterization was challenging, given the lack of a simple spectrophotometric assay. The nearly 7-fold difference between AUC and MST is, however, quite significant and perhaps should not be glossed over in the paper. The AUC measurement and ITC are much closer in agreement (0.1 vs. 0.35 micro molar). The fluorescent dye NST647 is used for MST, and could be significantly decreasing affinities by the modification of lysine residues. Perhaps the authors could perform an ITC titration with NST647-modified Rab, and see whether the affinity drops? The large difference leads to significant spread in the affinity and kinetic parameters, which has an impact on how to reconcile the differences in affinities between SopD2 and GtgE. Essentially, my concern is that MST is not providing a reliable affinity, relative to the other methods.

Also, the ITC data appears to be from a single measurement, and without a control (GtgE into buffer). A control seems warranted in this case, as there appear to be heat effects from dilution of ligand at the end of the titration. The errors shown (Fig 2C) seem to be from the fit to data of a single set of titrations. In this respect, ITC data is an 'outlier' relative to MST and AUC in not having been repeated for estimates of random error (rather than presumably systematic errors coming from plot fitting shown). I think it would be useful to resolve these issues with some control experiments and repetitions to narrow the spread of kinetic parameters.

We agree with the referee that the spread of determined Rab32:GtgE affinities is too high. We have initially considered the accuracy for the K_D (i.e. in the high nanomolar range) sufficient for the general interpretation of our findings. Because the true affinity of wildtype GtgE would be expected to be different from the determined K_D of the GtgE-C45A mutant, we were at first content with determining a K_D -range. However, stimulated by the referee we realized that indeed the K_D needs a more accurate determination.

Therefore, we now decided to focus on determining the complex affinity using isothermal titration calorimetry (ITC). Because ITC does not require labelling of neither of the proteins, the method provides a true measure of the protein-protein affinity. The K_D has now been determined multiple times and compared to the appropriate negative controls. The K_D value of the Rab32:GDP interaction with GtgE_{C45A} established at 96 nM +/- 31 nM. This is in very good agreement with the data obtained from analytical ultracentrifugation. Since the referee addressed concerns with respect to the validity of the MST data, we have decided to omit these from the final version of the manuscript.

The text of the manuscript has been changed to introduce these changes. Also, Figure 2 has been updated accordingly:

First, we subjected Rab32:GDP (20 μ M) to isothermal titration calorimetry (ITC) with GtgE_{C45A} (200 μ M), revealing a dissociation constant of $K_{D,ITC} = 96 \pm 31 \mu$ M (Fig. 2C). The determined stoichiometry confirms that GtgE and Rab32:GDP form a heterodimeric 1:1-complex. In

parallel, we utilized analytical ultracentrifugation (aUC) of Rab32:GDP (200 nM) covalently labeled at thiol groups with Atto488-Maleimide (Rab32:GDP_{fluor.}), and quantified complex formation with increasing concentrations of GtgE_{C45A} by difference in Rab32:GDP_{fluor.} sedimentation (Fig. 2D). Assuming a 1:1 binding stoichiometry, the K_D of the interaction is determined to $K_{D,aUC} = 118 \pm 29 \mu\text{M}$ which is in good agreement with the ITC-experiment.

Figure 2

Fig. 2. Inactive mutant GtgE_{C45A} has high affinity for inactive Rab32.

(A) Time dependent GtgE-mediated cleavage kinetic of Rab32:GDP with a SDS-PAGE gel shift assay. R: Rab32_{FL} (8 μM); G: GtgE_{WT} (8 nM). Spiked maltose binding protein (MBP, 2 μM) was used as internal quantification reference.

(B) Densitometric analysis of Rab32_{FL} bands from A fitted to single exponential function. The rate constant of the fit divided by the enzyme concentration yields the catalytic efficiency for GtgE.

(C) ITC of Rab32_{FL}:GDP (20 μM) titrated with 200 μM GtgE_{C45A}. Integrated heat peaks were fitted to a one-site-binding model yielding the binding stoichiometry (N), the enthalpy (ΔH), the entropy (ΔS), and the dissociation constant (K_D). Data are presented as means \pm SEM ($n = 3$).

Placing narrower spread for affinity is also important in understanding how Gtg could access Rab32(GDP) as it must compete with SopD, which has nM affinity for the Rab (Discussion). Also - why is it an evolutionary disadvantage to have a 'second secreted protein'? Is it simply the extra genetic load and the trouble of protein synthesis/secretion?

In addition, can the authors comment on how Gtg is able to cleave Rab32 when the affinity is at least 50-fold weaker, relative to GDI? Are there possibly separate pools of inactive Rab32 in infected cells? The authors vaguely hint at a 'cooperation' of GtgE and SopD2, but this should be clarified further. It is suggested in the discussion that **SopD2** could extract GtgE-cleaved Rab32, since the **SopD2**-Rab32 interaction is not significantly dependent on switch I. Is there experimental evidence that the GDI can interact with cleaved Rab32(GDP)? The suggestion that cleaved Rab32 could be extracted by **SopD2** from the SCV implies that GtgE can cleave membrane bound Rab32(GDP). Given the slow intrinsic GTPase activity of Rab32 (Fig 1C), how is the substrate for GtgE generated?

In the revised manuscript, we now describe the contribution of the individual proteins (i.e. GtgE, GDI, SopD2) in a more systematic manner. The referee is correct that the interaction order of Salmonella and human proteins with Rab32 was difficult to understand in the

previous version. We have therefore improved our model significantly. We hope that this provides better insights into the infectious cycle.

Due to the shortcomings in our initial description, the referee has probably been confused with SopD2 and GDI. We therefore speculate that the referee actually means GDI when he/she is referring to SopD2 in his/her comment (underlined in the comment above). In order to prevent further confusion we would like to state here that SopD2 is a Salmonella-GAP acting on active Rab32:GTP only and converts it into inactive Rab32:GDP, whereas GDI is a general Rab-solubilization factor that specifically binds to inactive prenylated Rab:GDP (here Rab32:GDP). Salmonella secretes two Rab32-targeting proteins during infection: The GAP SopD2 (specific for Rab32:GTP) and the protease GtgE (specific for Rab32:GDP). Due to the exclusive GDP-state specificity of GtgE we conclude that the presence of SopD2 is inevitable in order to convert the non-GtgE substrate Rab32:GTP into the GtgE-substrate Rab32:GDP.

The referee raises the question whether there are separate pools of inactive Rab32 in the cell and suggests that we clarify the cooperation between GtgE and SopD2 in inactivating the Rab protein. We therefore consider the following scenarios:

- A) GtgE targets Rab32:GDP produced after SopD2-mediated Rab32:GTP inactivation.
- B) GtgE cleaves Rab32:GDP produced by spontaneous dissociation from the complex with GDI.

We hypothesize that only the option A is realized for the following reasons (please also refer to the illustration below (Supplementary Fig. 19) for a graphical representation of the two scenarios):

I) As mentioned by the referee, the affinity of GDI for Rab32:GDP is about 50 fold higher than for GtgE. Hence, GtgE can only weakly compete with GDI for Rab32-binding. In addition, since GDI and GtgE occupy overlapping binding sites on Rab32, the protease can only process the GTPase once GDI has spontaneously dissociated at its intrinsic rate. Because of the slow rate of spontaneous dissociation¹, the Rab32:GDI complex disassembly is rate limiting for the subsequent proteolytic step by GtgE. In conclusion, direct targeting of the GDI-bound pool of Rab32:GDP is highly unlikely and would proceed at very slow rates.

Alternatively, Rab32 may be dissociated from GDI physiologically by the action of a corresponding GEF (e.g. BLOC3²). However, this would lead to the creation of a pool of membrane-bound Rab32:GTP. In the absence of a suitable GAP such as SopD2 this would provide a dead-end in the context of Salmonella infection since the GTP-bound form of Rab32 is no substrate to GtgE.

II) The recruitment of Rab-proteins to membrane is dependent on prior activation by GEFs, leading to the production of GTP-loaded Rabs on the membrane surface³. It is therefore reasonable to assume that Rab32 is first recruited to the SCV-membrane by an appropriate GEF (e.g. BLOC3)², leading to the creation of a pool of active membrane bound Rab32:GTP. Since Rab32:GTP is no substrate to GtgE, the GAP SopD2 ensures conversion of Rab32 into the GDP-bound state. This membrane-bound fraction of inactive Rab32:GDP is now targeted by GtgE leading to catalytically efficient cleavage.

In conclusion, SopD2 leads to a rapid production Rab32:GDP and thereby produces the GtgE-substrates (scenario A, Supplementary Fig. 19A). In this manner, cleaved GtgE is generated by a series of fast enzymatic conversion. In the absence of SopD2, however, Rab32-cleavage would be very slow due to the dependence on the spontaneous GDI-dissociation (scenario B, Supplementary Fig. 19B).

In order to make these coherences more comprehensible, we have now provided a schematic model to the supplement of the manuscript that explains the cooperation of SopD2 and GtgE (Supplementary Fig. 19 A-B). Furthermore, we have supplemented the manuscript with experiments demonstrating that catalytic quantities of SopD2 quickly convert Rab32:GTP into a GtgE-substrate (Supplementary Fig. 3). In addition, we have changed the main text into the following:

We therefore wondered whether GtgE cooperates with other enzymes in order to convert the Rab-proteins into the inactive form as required for proteolysis. Interestingly, *Salmonella* secretes the bacterial Rab32-GAP SopD2 that would be able to produce Rab32:GDP from Rab32:GTP¹⁵. Indeed, proteolysis only happens *in vitro* when SopD2 is added catalytically to the reaction between Rab32:GTP and GtgE (Supplementary Fig. 3). The rate of GtgE-mediated cleavage for Rab32:GTP in the presence of SopD2 was indistinguishable from the reaction of Rab32:GDP in the absence of this GAP. This indicates a cooperative deactivation mechanism of SopD2 with GtgE for effective Rab32-cleavage.

Supplementary Figure 3

Supplementary Fig. 3. Dependence of Rab32-cleavage by GtgE on SopD2 mediated GTPase activation.

(A) SopD2 interference with GtgE mediated proteolytic Rab32 cleavage over time analysed by Coomassie-stained SDS-PAGE. Rab32_{FL} (R₃₂; 8 μM) was challenged with GtgE (G_{WT}; 8 nM) and/or SopD2 (S; 80 nM). White triangle: MBP standard; black arrow: Rab32_{FL}; red arrow: cleavage product Rab32₆₀₋₂₂₅.

(B) Densitometric quantification of the time-dependent decrease of Rab32_{FL} bands from A. Signal intensity was normalized for the internal standard MBP. Data was fitted to a single exponential function to yield the observed rate constants (k_{obs}) (bottom right).

(C) Bar graph of the determined rate constant from B.

Supplementary Figure 19

Supplementary Fig. 19. Mechanistic model of the dependence of GtgE-mediated Rab32-cleavage on SopD2. The cooperation of SopD2 and GtgE allows rapid Rab32-inactivation since the proteolysis is a result of a series of fast enzymatic conversions. In contrast, in the absence of SopD2, GtgE would need to compete with GDI for binding to Rab32:GDP. Because GDI binds to GDP-loaded Rab proteins strongly, Rab32-cleavage by GtgE would be prohibited.

(A) Cooperative model of SopD2 and GtgE activity for processing active Rab32. For legend see Supplementary Fig. 19B. Rab32 is recruited to the membrane by a corresponding GEF (BLOC-3). The resulting SCV-bound GTP-loaded GTPase is then rapidly converted by the action of the GAP SopD2. Eventually, Rab32:GDP is cleaved quickly by GtgE.

(B) Model of GtgE mediated-cleavage of active Rab32 or GDI-bound inactive Rab32 in the absence of SopD2. The Rab-protein may be recruited to the membrane by a GEF. However, the resulting Rab32:GTP is no substrate of GtgE, prohibiting Rab32-cleavage via this pathway. Alternatively, GtgE could hypothetically target cytosolic Rab32:GDP in complex with GDI. The high affinity of the Rab:GDI-complex, however, would exclude efficient turnover of Rab32:GDP by GtgE. Thus, the absence of SopD2 would make the time-efficient cleavage of Rab32 at the SCV very unlikely. GEF: Guanine nucleotide exchange factor.

Furthermore, the referee asks whether the cleaved Rab32 is still able to bind to GDI and to be extracted from the membrane since we have mentioned this in our discussion. Furthermore, the referee requested if experimental data are available to support this speculation. As of now, we cannot support this hypothesis with appropriate results. We have therefore indicated in the main text that this interpretation is highly speculative. A profound analysis of the GtgE-cleavage consequences for the interactions of Rab32 with regulators and effectors will be part of a follow-up research project.

In summary, the findings from in vitro assays and the crystal structure need better connections to biology, i.e., how does GtgE contribute to Salmonella infections? As it stands, the manuscript certainly gives a better understanding of the in vitro catalytic activity of the

enzyme. Although clearly interesting and novel, the main shortcoming of the paper is how that activity fits into the life cycle and host range of Salmonella infection.

Stimulated by the referees comment we realized that the connection between *in vitro* data and *in vivo* consequences needs clarification and optimization in comprehensibility. We have therefore significantly improved the logic of our *in vitro* data interpretations and their relevance for the Salmonella infection mechanism. These and other changes mentioned before now contribute mechanistically to the molecular events during infections with respect of GtgE. The following changes have been included into the discussion that focus on the interplay between SopD2 and GtgE and the relevance for the Rab:GDP-specificity of the protease:

In contrast, the combination of SopD2 and GtgE Salmonella at the SCV membrane during infection guarantees fast Rab32-inactivation as a results of a series of rapid enzymatic steps (Supplementary Fig. 19A): Rab32:GDP is likely recruited to the SCV via a GEF (BLOC-3) from its cytosolic complex with GDI promptly. The resulting pool of active Rab32:GTP is then immediately targeted by the GAP SopD2, leading to GTP-hydrolysis and the production of membrane-localized Rab32:GDP. Finally, GtgE is attacking Rab32:GDP at the SCV and the proteolyzed GTPase is formed quickly. Therefore, the cooperation of SopD2 and GtgE may overall be the most efficient way for quantitative Rab32-proteolysis (Supplementary Fig. 19A). An efficient Rab-inactivation therefore appears to be crucial for the progressing SCV maturation.

The other main issue that I have with the manuscript concerns the MD work. The authors start with the pre-formed complex, but as comparison, substitute GTP for GDP to evaluate specificity. However, Rab32(GTP) does NOT form a complex with GtgE, therefore its complex with GtgE (at start of simulation) is artificial. Furthermore, Rab32(GTP) conformation is distinct from Rab32(GDP), and so this experimental strategy seems highly contrived. The simulation does appear to succeed in identifying a determinant of GTP specificity, and is subsequently supported by mutagenesis and enzymatic assays. However, this positive result does NOT obscure the fact that the MD strategy is poorly designed. The current simulation may show why GTP destabilizes the Rab32 (starting from a GDP state), but is NOT designed to provide insight into pre-binding preferences for GDP vs. GTP conformations. The key question of pre-binding compatibility (GDP vs. GTP) cannot be addressed with this kind of simulation.

I can outline several suggestions:

- at the very least, the authors must state the shortcomings in this simulation and explain clearly why they chose this strategy
- the structure of Rab32(GTP) is known. An alternative simulation is to 'steer' the complex towards the GTP conformation (but keep the GDP in complex), and see how the system reacts. Still not ideal, but there is less chance for the simulation 'exploding'
- detach the complex a small distance, and steer either Rab32(GDP) or Rab32(GTP) toward the known structure of the complex. This would, in principle, provide the best simulation of the binding preferences. However, this would require major investment in computing time and is more 'artificial' and likely to fail.
- it might be useful to validate simulations by looking at the binding affinity of the F88G mutation of Rab32 in the GTP state. Comparisons of GTP-bound F88G would provide significant support for the hub3/spike contribution to specificity. Presumably it would be a weak interaction, but certainly more than wild-type Rab32(GTP). Clearly the mutant is enabling cleavage of the GTP conformation (Fig 5F), therefore binding affinity is significant

and can be quantified. The experiment would provide considerable support for the model for GDP selectivity.

We thank the referee for the insightful suggestions. Based on the reviewer's comment, we realize that our computational approach needed further clarifications.

We first indeed considered to simulate the formation of the Rab32:GtgE complexes by comparing both nucleotide states (i.e. GDP and GTP) side-by-side by application of extensive free energy simulations approaches. However, this methodology would have required considerable computing resources with limited success chances. This estimation concurs with the reviewer comment.

Importantly, in the two crystal structures with bound GppCp (PDB ID: 4CYM) and GDP (crystallized by our groups), respectively, Rab32 is in complex with two different proteins: In the GDP-bound state, Rab32 was crystallized by us in complex with GtgE, while in the GppCp-bound state, Rab32 was crystallized in complex with VARP-ANKRD1. This results in different conformations of Rab32 in the interaction site with the counter protein, and replacement of structure of the GppCp-bound Rab32 into the Rab32:GtgE results in steric clashes (see figure below). It was therefore not possible to use the crystal structure with GppCp as a reference structure for initiation of our simulations. This has been clarified in the revised text and with an additional supplementary figure (Supplementary Fig. 16).

Supplementary Figure 16

Supplementary Fig. 16. Models of GDP-bound (top panel) and a hypothetical GTP-bound Rab32 (bottom panel) in complex with wild type GtgE as starting points for atomistic molecular dynamics simulations. The GDP-bound Rab:GtgE complex is constituted based on the presented crystal structure from this work. The hypothetical GTP-bound complex

structure is based on the active state Rab32 in complex with GppCH₂p and its effector binding domain VARP-ANKRD1 (PDB ID: 4CYM). Only the hypothetical Rab32:GTP:GtgE results in steric conflicts along the protein-protein interface particularly in the Rab switch I region with the spike 2 loop in GtgE. Both complexes are additionally shown 45° tilted on the y-axis.

Since we were interested in the amino acid determinants that would be incompatible with GtgE binding to active Rab32:GTP, we applied a different strategy: We made use of the fact that all small GTPases can form complexes with their binding partners in both activity states. Even effectors that preferentially interact with active G-proteins, do bind to the inactive state albeit with 1000-fold lower affinity. The same arguments apply to GDI that preferentially binds to inactive GTPases but can also interact very weakly with the active forms. Thus, these low-affinity complexes are produced but have a very low likelihood of occurrence. Therefore, instead of addressing the issue why GtgE cannot recognize active Rab32:GTP, we probed which determinants would contribute most to dissociating the Rab32:GTP:GtgE-complex once formed. Hence, the MD-trajectories calculated in our work are essentially representing the initial molecular steps of GtgE dissociating from Rab32:GTP. We are convinced that this type of inverted analysis is actually reflecting the binding equilibrium more realistically: It is very common to protein-protein-interaction that the affinity is mainly determined by the rate of complex dissociation (i.e. k_{off}) rather than by the second order rate constant of complex association (i.e. k_{on}). While k_{on} usually is in a narrow range of 10^6 - 10^7 M⁻¹s⁻¹, k_{off} can span many orders of magnitude, e.g. 10^{-5} - 10^3 M⁻¹s⁻¹. Thus, the affinity of a complex is not determined by the rate of the encounter of the proteins but rather by the rate of complex disintegration. In our approach, we have followed this logic and simulated the complex dissociation by purposefully placing GTP instead of GDP in the nucleotide binding pocket of Rab32.

We also wish to clarify that the F88G-mutation was identified based on our MD simulations of the dissociation trajectory (not vice versa). Further analysis of the simulations data to elucidate an exact dissociation mechanism would indeed be interesting but is outside the scope of this work. This has also been better clarified in the revised text.

We hope that we could convince the referee about the correctness of using the strategy outlined above, but we realize that this concept might not be self-understood. We have therefore improved our manuscript accordingly. The following changes have been made to the manuscript:

In order to obtain additional insights into the nucleotide-state specificity of GtgE, we performed atomistic molecular dynamics (MD) simulations. We first considered calculating the structure of a hypothetical complex between active Rab32:GTP and GtgE. Therefore, the only currently available structure of active Rab32 bound to the non-hydrolysable GTP-analogue GppCH₂p was taken from its complex with the effector protein VARP (PDB ID: 4CYM)¹⁷. Replacement of the structure of the GppCH₂p-bound Rab32 from its complex with VARP resulted in steric clashes of Rab32 with GtgE despite the global structure of Rab32 closely resemble each other in the two X-ray structures (Supplementary Fig. 16). Instead of probing the formation of Rab32:GTP:GtgE complex by extensive free energy calculations, we computationally replaced GDP with GTP in the nucleotide binding pocket of Rab32 and probed the structural determinants for GtgE-dissociation from the complex. This approach is justified since small GTPases can form complexes with their binding partners in both activity states albeit with greatly differing affinities²³. It is common, however, that protein-protein-affinities are mainly determined by the rate of complex dissociation (k_{off} , 10^{-5} - 10^3 M⁻¹s⁻¹)

rather than by the second order rate constant of complex association (k_{on} , 10^6 - 10^7 $M^{-1}s^{-1}$). The MD-simulation therefore mimics the initial molecular steps of GtgE dissociating from Rab32:GTP, and aim to probe determinants that would contribute most to dissociating the Rab32:GTP:GtgE-complex once formed. These simulations do not therefore address the rate of complex association, since they would require extensive free energy simulations, and are outside the scope of the present study.

To clarify the relevance of the computationally predicted F88G_R mutant:

In order to probe the function of F88_R in Rab32, we replaced the residue with a glycine (F88G_R), and re-initiated the MD simulations. Interestingly, the MD simulation of the F88G_R demonstrate that the GTP- and GDP-states behaves more similar in this mutant (Supplementary Fig. 18C). Therefore, to validate the significance of hub 3 for the GtgE selectivity towards GDP over GTP, we produced the Rab32 F88G_R-mutant also experimentally.

A few other comments and suggestions:

- quality and coloring of molecules (ribbons, Fig 3) need revisions, difficult to tell Rab/GtgE apart at the interface. The interior surface of Gtg ribbon is the same color as Rab32 (grey), so the colors run into each other. Similar problems with 1B, 1C (superpositions) - ribbons should be a single color (both surfaces), and color choice should contrast better to identify molecules.

According to the referee's suggestion we have standardized the cartoon coloring to a uniform representation of inner and outer ribbon surface in Figure 3A-C for better clarity. Furthermore, we increased the contrast between the colors chosen for super positioning of Rab32 (orange/grey with magenta highlight) and GtgE structures (dark blue/cyan) respectively (Figure 3B-C).

Figure 3A-C

- similar objections to Fig 5B, very difficult to see superpositions, the ribbons (interior/exterior) must be uniform color to reduce complexity

According to the referee's suggestion, we have standardized the cartoon coloring to a uniform representation of inner and outer ribbon surface in Figure 5B and higher color contrast in Figure 5A-B. The following updated figure has been added to the manuscript.

Figure 5A-B

- grammar - rephrase 'would result in the creation of active Rab32:GTP that is no GtgE-substrate' to 'would result in the creation of active Rab32:GTP that is not a GtgE-substrate'

The grammar error was corrected according to the referee's suggestion:

... would result in the creation of active Rab32:GTP that is not a GtgE-substrate ...-

typos and technical issues:

- spelling 'Rab32-clevege' to 'Rab32-cleavage', in 'GtgE has high affinity...' section
- typo - 'an hydrophobic => a hydrophobic cluster' during structural description of interface
- 'obvisouly the recruitment of Rab32 to the SCV'. Also, why is Rab32 recruitment to SCV 'obviously' important?

Spelling errors and typos have been corrected as suggested by the referee. We also have addressed the comment on the "obvious importance of Rab32 recruitment to the SCV" raised by the referee. We realized that this statement may be confusing and have therefore rephrased the corresponding section of the discussion as follows:

...We hypothesize, however, that the presence of SopD2 ensures time-efficient access to GDP-loaded Rab32 since the latter would be blocked by tightly binding to GDI. We therefore suggest the following order of events: The Rab-protein is targeted to and activated at the membrane by a corresponding GEF. ...

- the average B-factors are very high for the full-length GtgE complex, relative to other structures at 3Å resolution. Can the authors comment on this? In particular, the nucleotide average is 114, it might be useful to show the density in this region since it is somewhat irregular to have such high B-factors

Indeed, the B-factor of the GDP-molecule in the full-length Rab32:GtgE-complex (PDB ID 5OED, space group P6₅22, 2.9 Å resolution) is increased compared to the protein residues. We suggest that crystallization conditions and/or crystal packing might cause a reduced occupancy of the nucleotide compared to the structure of the truncated Rab32₁₈₋₂₀₁:GDP:GtgE₂₁₋₂₁₄ complex (PDB ID 5OEC, space group P2₁2₁2₁, 2.3 Å resolution). We have used an occupancy of 1.0 for both GDP molecules in our refinements, therefore resulting in an increased temperature factor of the former. We agree with the referee that the average B-factors are high for the full-length GtgE complex relative to other structures at 3 Å resolution, and therefore have included an additional figure displaying the 2F_O-F_C electron-density maps of GDP-molecules for both complex structures, which have been excluded prior to phase calculations (in the Supplementary Fig. 9). The omit-density maps nicely display defined electron densities for the GDP molecules, respectively, and the superposition of both sections reveal a perfect match of both structures in this region. The following changes in the text have been made.

The average B-factor of the full length complex is higher than for the truncated complex, but electron density is well defined for both structures (Supplementary Fig. 9). Therefore, the truncated Rab32₁₈₋₂₀₁:GDP:GtgE_{21-214,C45A}-complex is discussed only and referred to as Rab32:GDP:GtgE_{C45A} hereafter.

Supplementary Figure 9

Supplementary Fig. 9. Comparison of the electron density of GDP in the full length and truncated Rab32:GtgE_{C45A} complex structures.

(A) Sections of the P-loop, the GDP-molecule and the magnesium ion of Rab32₁₈₋₂₀₁:GDP:GtgE_{21-214,C45A} (left panel, PDB ID: 5OEC, space group P2₁2₁2₁, 2.3 Å resolution) and Rab32_{FL}:GDP:GtgE_{FL,C45A} (right panel, PDB ID: 5OED, space group P6₅22, 2.9 Å resolution), respectively. The 2F_O-F_C electron-density maps depict the GDP nucleotides, which have been excluded prior to phase calculations (blue mesh, contour level at 1σ).

(B) Structural superposition of both crystal structures revealing a perfect match of both complexes within this region.

Reviewer #2 (Remarks to the Author):

The manuscript by Wachtel et al. details comprehensive biochemical and structural studies and complementary molecular dynamics simulations that mechanistically demonstrate how a critical bacterial virulence factor binds to and cleaves host Rab GTPases. The bacterial pathogen *Salmonella typhimurium* expresses two virulence factors a RabGAP (SopD2) and a cysteine protease known as GtgE. Together these factors serve to inactivate host Rab GTPases that support pathogen elimination through phagolysosomal fusion. In particular, the GtgE protease supports intracellular survival and replication through the cleavage of host Rab GTPases. GtgE is essential to the expanded host range of *Salmonella typhimurium* relative to *S. typhi*. Even while two prior structural studies of proteolytic fragments of GtgE identified it as a cysteine protease, details of Rab substrate interaction and catalytic specificity remained unclear due to the use of peptide substrates rather than intact Rab GTPase proteins.

The present determination of the mechanistic underpinnings of GtgE proteolysis through biochemical and structural analyses represents an important advance. The work clearly identifies GDP-bound Rab GTPases (Rab29, Rab32 and Rab38) as the substrates of full length GtgE. The work also clarifies the additive virulence benefit to *Salmonella* strains that express a RabGAP to increase the Rab-GDP substrate pool, and GtgE to promote Rab proteolysis. It is speculated that even though GtgE cleavage does not destabilize the Rab it prevents further GEF mediated activation of the cleaved GTPase. Additionally, the study also identifies a phenylalanine in the switch 2 domain of Rab32 as critical to activity-state discrimination, which may be generalizable to GDI protein selectivity for Rab-GDP.

Overall, the work is highly compelling, well controlled and statistically validated. Molecular dynamics predictions are used to augment mutational analyses and to explicitly test specific predictions. The findings are groundbreaking, considered of significant interest to the Nature Communications readership and therefore highly recommended pending clarification of the text regarding which experiments were performed with FL vs. mutant proteins and interpretations of the mutant analyses in the context of non-binding Rab23.

Specific Comments

Methods and results sections indicate that both FL and truncated proteins were purified and used for structure determinations. In the Results section and for each figure indicate clearly which proteins were used, e.g. Fig. 4 shows truncated protein structure, yet text refers to wild-type GtgE on line 315. Under DNA cloning (lines 564-579) include the methods for construction of the truncated proteins.

Indeed, the discrimination between FL and truncated proteins required clarification. The cloning and application of truncated or full-length versions of Rab32 and GtgE are clarified in the new manuscript version now. The truncated proteins were used only in the crystallization and structure determination process. All activity and binding assay were conducted with full length Rab32 or GtgE, including the indicated mutations, respectively. The following changes have therefore been made:

...Additionally, truncated Rab32₁₈₋₂₀₁ and GtgE_{21-214,C45A} were generated in the pMAL vector as described above and used for protein crystallization and structural studies, only. ...

Additionally, we have specified throughout the result section and the methods which proteins have been used.

The work considers GtgE interactions with Rab32, Rab38 and Rab29 as contrasted to non-binding Rab23. However, there is a missed opportunity in the analyses to explain the selectivity for binding to particular Rabs and which amino acids might strengthen affinity. For example, different mutants are used for yeast two hybrid studies to measure altered Rab32-GtgE binding relative to GtgE proteolysis, e.g. R87A and L64A mutants were used for binding and R87E and L64E were used for proteolysis studies. When comparing the sequence alignments of the 4 rabs in Supplementary Fig. 1 it appears that there may be interesting differences relative to Rab23 in the beta2 and beta3 sequences that could account for the binding specificity of GtgE for Rab32, Rab38, and Rab29. As presented, because the alanine mutations don't block binding, one cannot surmise if the glutamate substitutions block binding and cleavage or just proteolytic activity. In line 325 it is stated that specific amino acids cause a growth defect confirming role in Rab32:GtgE complex formation. However, all of these amino acids are present in Rab23 so why doesn't it bind GtgE? Similarly line 338, W80 and F62 are conserved in Rab23. Perhaps there are some key amino acids in the beta2-beta3 regions that may play a role? Line 348—need to more clearly distinguish the mutants that are involved in substrate recognition.

The referee raises concerns of missing the opportunity for obtaining insights into the Rab-binding profile of GtgE. It has been shown previously that GtgE can bind to Rab32, Rab38, and Rab29, but not to the closest homolog Rab23. The referee asks how the discrimination between Rab32/29/38 and Rab23 is achieved on a molecular level. To address this issue we identified four residues in Rab32 that are involved in the GtgE interaction and have significant different chemical properties in Rab23. These potential key amino acids are spread over the whole interacting area in three hot spots located in the switch I (K40A), the beta2 (interswitch; E48L and Q50V), and the switch II region (E70R) – referring to the Rab23 sequence numbering (see Supplementary Fig. 1). We have mutated the four positions in Rab23 into the corresponding Rab32 amino acid and studied their impact on the GtgE-mediated proteolytic cleavage. To our surprise even after long incubation the GDP bound Rab23 mutant was not accepted as substrate by GtgE (Supplementary Fig. 14). This result indicates that the GtgE-mediated substrate selection is based on multiple factors and can not be elucidated by the exchange of specific amino acid side chains. We introduced this aspect also in the main text and the supplement.

The Rab32:GDP:GtgE complex structure was also used as a basis to obtain insights into the Rab-selectivity. Based on the amino acid sequence alignment between the GtgE-substrates Rab32, 29, 38 and the non-GtgE-substrate Rab23 we identified four amino acids that are located in the protein-protein interface and may be incompatible with binding of Rab23 to the protease (Supplementary Fig. 1). We then tested whether we could convert Rab23 into a GtgE-substrate by introducing the Rab32 analogues amino acids K40A, E48L, Q50V, and E70R *in vitro*. However, this Rab23 mutant was not cleaved by GtgE indicating a multifactorial selection mode that cannot be reduced to a few specific amino acid side chains (Supplementary Fig. 14).

Supplementary Figure S14

Supplementary Fig. 14. Exploration of the GtgE Rab-substrate specificity with a selected Rab23 mutant. Choice of mutation positions was guided by sequence alignment and interaction site analysis from the Rab32:GtgE-complex (see Supplementary Fig. 1).

(A) Structural superposition of the Rab23:GDP (orange) and Rab32:GDP (grey) in complex with GtgE_{C45A} (blue, transparent) depicted as cartoon. Mutations in Rab23 to the corresponding amino acid from Rab32 were identified in three hot spots highlighted in red circles: switch I (K40A); interswitch (E48L; Q50V); switch II (E70R). Highlighted as red spheres: C_α-atoms of respective mutations, sticks: GDP, magenta: switch regions, green sphere: Mg²⁺-ion.

(B) Time dependent GtgE proteolysis gel shift assay of Rab32:GDP, Rab23:GDP, and Rab23:GDP mutant (K40A; E48L; Q50V; E70R). GtgE is unable to cleave Rab23 or Rab23-mutant containing respective Rab32-mutations, indicating that further structural determinants contribute to the Rab-specificity of the protease. Abbreviations: GtgE WT (GWT, 8 nM), respective GDP-bound Rab (RGDP, all 8 μM) without GtgE added.

The referee also asked to “*more clearly distinguish the mutants that are involved in substrate recognition.*” We interpreted this statement as a wish to clarify the contributions of the chosen amino acids in the complex interface. We have therefore provided the following text additions thereby focusing on the amino acid mutations that have not been discussed in the previous manuscript version:

... The mutants K43A_R and R93A_R abrogate salt bridge formation with E148_G and D182_G, respectively, hence confirming the relevance of the corresponding ionic interaction for complex formation. In addition, the decrease in activity for the R87E_R mutant supports the importance of the arginine side chain in binding to GtgE spike 3 via a backbone H-bridge with K194_G. The mutant A56K_R presumably causes a steric clashes with the side chains of L150_G and P143_G that form the basis of the important spike 2 loop of GtgE (Fig. 3E) thereby corroborating the relevance of this region for recognition of Rab32 by GtgE. ...

... The importance of the D61_R-R142_G salt bridge as a central interaction site mentioned before is once more confirmed using the R142A_G mutation. ...

Minor Corrections

Introduction: line 93 change 'deactivation' to inactivation

Fig. 1 line 146 change 'means of a technical replica' to means of technical replicates

Line 161 Rab32-cleavage educts should be Rab32-cleavage products

Line 164 To this purposes should be For this purpose

Line 171 change to: ...would require Michaelis-Menten-kinetics to be conducted at

Line 188: change to: ..determined to be Kd

Line 505: change to that is not a GtgE-substrate

Line 536: change to: despite of Rab32

Line 544 and 545: be more clear that this is speculative

Line 567: change to synthesized

Line 598: change to subjected to Tobacco Etch

Line 612: instead of one equivalent state that 8 nM GtgE was used

Supplemental section: Change Varients to Variants in the A and B headings of Supplemental Figure S10

Line 94: Change S7 to S5

The terminology 'deactivation' was consistently changed for 'inactivation' in the context of the Rab-activity state regulation. All further suggested corrections have been addressed and were implemented according to the referee's recommendations.

References mentioned in the referee responses

1. Wu YW, Oesterlin LK, Tan KT, Waldmann H, Alexandrov K, Goody RS. Membrane targeting mechanism of Rab GTPases elucidated by semisynthetic protein probes. *Nat Chem Biol* **6**, 534-540 (2010).
2. Gerondopoulos A, Langemeyer L, Liang JR, Linford A, Barr FA. BLOC-3 Mutated in Hermansky-Pudlak Syndrome Is a Rab32/38 Guanine Nucleotide Exchange Factor. *Curr Biol* **22**, 2135-2139 (2012).
3. Blümer J, *et al.* RabGEFs are a major determinant for specific Rab membrane targeting. *J Cell Biol* **200**, 287-300 (2013).
4. Hesketh GG, *et al.* VARP Is Recruited on to Endosomes by Direct Interaction with Retromer, Where Together They Function in Export to the Cell Surface. *Dev Cell* **29**, 591-606 (2014).

Reviewers' Comments:

Reviewer #1 (Remarks to the Author):

The authors have performed additional experiments and clarified the role of the various Salmonella proteins in the subversion of Rab GTPases.

I appreciate the additional paragraph explaining the co-operation between SopD and GtgE. The KD for the interaction between GtgE and Rab32 is also consistent with two independent methods. The electron density is also clear for GDP, despite high B-factors. Also, the MD simulations are more clearly justified in the revised manuscript.

The revised manuscript has addressed my concerns and is now suitable for publication.

a few minor revisions:

- suppl fig 9, legend, change 'perfect match' to 'close match'
- line 378-379 - I would change 'analysis confirms' to 'analysis suggests', since it is debatable whether MD work is considered legitimate experimental data
- line 394, change 'preparatively loaded' to 'purified'
- line 449, presumably the phrase should be 'since it is dependent' rather than 'since the dependence'
- suppl Fig 19, legend B, should read 'the resulting Rab32:GTP is not a substrate of...'
- suppl Fig 19 - change 'cytosole' to 'cytosol'

It is a little unusual to use the term 'Rab-proteins' in the title, I would change to the conventional term 'Rab GTPases'.

Finally, I would recommend that Supplementary Fig 19 (panel A) become Fig. 6 of the main paper, it is a useful guide for non-specialists and nicely summarizes the roles of Salmonella proteins. I am not sure whether panel B is necessary. Is there a strain of Salmonella that has GtgE, but not SopD? If not, then panel B of Suppl Fig 19 is a hypothetical situation and is misleading to readers. The consequence of GtgE activity in the case that SopD is limiting or lacking could be addressed using panel A.

Reviewer #2 (Remarks to the Author):

I am fully satisfied with the authors' responses to the prior review. The new experimental measurements of the Kd of Rab32:GTGE interaction and the demonstration that GTGE is unable

to inactivate Rab32 through cleavage in the absence of SOPD2 offer important substantiation of the conclusions. The data together with text edits increase the rigor and improve the readability of the work. The data support a seminal mechanistic model for how the two bacterial factors enable Salmonella to set up a productive infection within the infected host cell.

Response to referees

The protease GtgE from Salmonella exclusively targets inactive Rab GTPases

Again, we would like to thank the referees for their time and their contributions. We are grateful for the additional comments, for the corrections of errors, and for further suggestions. We are now providing the final version of the manuscript. In addition, we have made use of the referee recommendations in order to improve the paper.

Reviewer comments: Black
Author responses: Blue
Text changes in the revised manuscript: Red

Reviewer #1 (Remarks to the Author):

The authors have performed additional experiments and clarified the role of the various Salmonella proteins in the subversion of Rab GTPases.
I appreciate the additional paragraph explaining the co-operation between SopD and GtgE. The KD for the interaction between GtgE and Rab32 is also consistent with two independent methods. The electron density is also clear for GDP, despite high B-factors. Also, the MD simulations are more clearly justified in the revised manuscript.
The revised manuscript has addressed my concerns and is now suitable for publication.
We thank the referee for the positive reception of our manuscript changes.

a few minor revisions:

The following comments were addressed according to the referee's recommendations.

- suppl fig 9, legend, change 'perfect match' to 'close match'
“(B) Structural superposition of both crystal structures revealing a close match of both complexes within this region.”
- line 378-379 - I would change 'analysis confirms' to 'analysis suggests', since it is debatable whether MD work is considered legitimate experimental data
“Consequently, the analysis suggests that Rab32 is binding to GtgE in the GDP-state but dissociates from the protease in the GTP-state.”
- line 394, change 'preparatively loaded' to 'purified'
“To this end, we purified the full length Rab32 wild type and the F88GR-mutant with GDP or GppNHp, ...”
- line 449, presumably the phrase should be 'since it is dependent' rather than 'since the dependence'
“The question arises as to why Salmonella has evolved with a GDP-state specific Rab-protease since it is dependent on a second secreted protein (i.e. SopD2).”
- suppl Fig 19, legend B, should read 'the resulting Rab32:GTP is not a substrate of...'
- suppl Fig 19 - change 'cytosole' to 'cytosol'
Both points have been addressed together with the relocation of Suppl. Figure 19A into the main text as Figure 6 (see below).

It is a little unusual to use the term 'Rab-proteins' in the title, I would change to the conventional term 'Rab GTPases'.

We have changed the title accordingly.

"The protease GtgE from Salmonella exclusively targets inactive Rab GTPases"

Finally, I would recommend that Supplementary Fig 19 (panel A) become Fig. 6 of the main paper, it is a useful guide for non-specialists and nicely summarizes the roles of Salmonella proteins. I am not sure whether panel B is necessary. Is there a strain of Salmonella that has GtgE, but not SopD? If not, then panel B of Suppl Fig 19 is a hypothetical situation and is misleading to readers. The consequence of GtgE activity in the case that SopD is limiting or lacking could be addressed using panel A.

We agree with the referee's doubts regarding a potential misunderstanding caused by Suppl. Fig. 19B as the absence of SopD2 in a GtgE containing Salmonella strain is not documented and therefore represents a hypothetical scenario. Hence, we introduce Supplementary Fig. 19 (panel A only) as new concluding Fig. 6 in the main text as suggested.

Figure 6

Fig. 6 Mechanistic model of the dependence of GtgE-mediated Rab32-cleavage on SopD2. The cooperation of SopD2 and GtgE allows rapid Rab32-inactivation since the proteolysis is a result of a series of fast enzymatic conversions. Rab32 is recruited to the membrane by a corresponding GEF (BLOC-3). The resulting SCV-bound GTP-loaded GTPase is then rapidly converted by the action of the GAP SopD2. Eventually, Rab32:GDP is cleaved quickly by GtgE. GEF: Guanine nucleotide exchange factor, GDI: GDP dissociation inhibitor.

Reviewer #2 (Remarks to the Author):

I am fully satisfied with the authors' responses to the prior review. The new experimental measurements of the Kd of Rab32:GTGE interaction and the demonstration that GTGE is unable to inactivate Rab32 through cleavage in the absence of SOPD2 offer important substantiation of the conclusions. The data together with text edits increase the rigor and improve the readability of the work. The data support a seminal mechanistic model for how the two bacterial factors enable Salmonella to set up a productive infection within the infected host cell.

We thank the referee for the positive evaluation of our revised manuscript.